# K-FRAMES: SCENE-DRIVEN ANY-k KEYFRAME SELECTION FOR LONG VIDEO UNDERSTANDING

## ABSTRACT

Multimodal Large Language Models (MLLMs) have demonstrated significant capabilities in image understanding, but long-video are constrained by context windows and computational cost. Uniform frame sampling often leads to substantial information loss. Meanwhile existing keyframe selection methods such as text-frame retrieval or RL-based frame optimization typically yield sparse and temporally disjointed frames, overlooking scene continuity and lacking flexibility for multi-scale frame selection. To address these limitations, we introduce K-frames, a novel paradigm for scene-driven keyframe selection that preserves temporal continuity. Instead of selecting individual frames, K-frames predicts semantically coherent, query-relevant clips, which enables any-k keyframes selection to meet diverse user budgets. To achieve this approach, we first introduce PeakClips, a dataset of 200K video highlights conditioned by query. Building on this dataset, K-frames learns clip2frame selection using a three-stage progressive curriculum. It involves two Supervised Fine-Tuning stages for temporal grounding and key-clip perception, followed by a Reinforcement Learning stage that directly optimizes the scene-driven prediction policy for downstream task without further annotations. Extensive experiments on major long-video understanding benchmarks demonstrate that K-frames provides an effective, interpretable, and plug-and-play solution for keyframe selection at various scales. Our dataset and model will be available.

## 1 INTRODUCTION

Recent progress in Multimodal Large Language Models (MLLMs) (Bai et al., 2025; Wang et al., 2025) has come from coupling Large Language Models (LLMs) with vision encoders via a cross-modal projector that maps visual features into the language token space. This design enables unified, instruction-following multimodal reasoning across diverse text-image tasks. However, extending these models from image to video remains challenging. As treating a video as a sequence of frames greatly increases the number of visual tokens, especially for long videos. On the one hand, finite context windows cannot accommodate all video frames. On the other hand, the quadratic computational complexity of standard Transformer attention (Vaswani et al., 2017) makes longer inputs dramatically more expensive in computation and in token-metered API usage. Therefore, frame downsampling is practically necessary for video inputs.

Current MLLMs typically process videos via uniform frame sampling. But for long videos, the challenge is that sampling only a small subset of frames risks a critical loss of context, highlighting the need for keyframe selection. Existing methodologies for keyframe selection are predominantly categorized into two paradigms: text-frame retrieval and Reinforcement Learning (RL)-based optimization. The former computes the similarity of frames and text query to rank frames (Tang et al., 2025), treating video as independent images. This neglects temporal context and struggles with instruction-heavy or compositional queries. The latter, RL-based methods, optimize frame subsets for downstream objectives. But the resulting selections are typically sparse, which harms scene continuity, thereby degrading video understanding performance. And it also fails to accommodate personalized user budgets due to the lack of flexibility for multi-scale selection.

To address these limitations, we propose K-frames, a query-conditioned and interpretable paradigm that reframes keyframe selection as clip2frame prediction. Instead of selecting isolated frames,

**Query:** What the correct order for the events in video?
(1) The man in red cap stands outside a barbershop talking;
(2) The man pretends to be asleep during his haircut;
(3) The man points out the camera and explains it to barber;
(4) The man appears to fall out of the chair.

A.2→1→3→4  B.3→2→1→4
C.1→2→4→3  D.1→3→2→4

**Downstream Prediction:** C. 1→2→4→3 ✅

**K-frames Output**
<time>7-13, P2,</time><reason>It shows the first event: the man in the red cap standing outside the barbershop talking;</reason>
<time>13-35, P1,</time><reason>It shows the man pretending to be asleep during his haircut, which is the second event. |
It further shows the man pointing out the cameras to the barber, which is the third event;</reason>
<time>35-52, P2,</time><reason>It shows the man appearing to fall out of the chair, which is the fourth event ;</reason>

Figure 1: Visualization of our K-frames paradigm. Unlike uniform sampling (UNI), our model first predicts query-relevant key clips along the video timeline, assigning them importance levels of P1 (top-priority) or P2 (secondary-priority). Keyframes are then selected based on these key clips.

K-frames first localizes semantically coherent, temporally contiguous clips aligned with the query, and then selects any-k keyframes based on those clips. As illustrated in Figure 1, this clip-first design preserves scene continuity, focuses computation on informative regions, making the selection process interpretable. As a model-agnostic front-end, K-frames enhances the efficiency and performance of existing MLLMs in long video understanding with no modifications to their architecture.

The main challenge in scene-driven keyframe selection is the lack of scene-level relevance annotations. To close this gap, we construct a new dataset, **PeakClips**, with hierarchical captions and detailed video highlight annotations. PeakClips is built via a three-stage pipeline: (1) scene segmentation partitions videos into scene-aware temporal units based on changes in visual content; (2) hierarchical captioning at the scene/chapter/video levels supplies multi-granular descriptions that link local scene to the global narrative; and (3) LLM-guided relevance scoring aligns scenes with the query through Gemini 2.5 Pro (Comanici et al., 2025), and using frame–query similarity further refines relevance score to the frame level. By annotating these scenes, we ultimately aim to supply keyframe selection, temporal localization, and hierarchical understanding in long-term video.

Building on the PeakClips dataset, we employ a three-stage progressive curriculum to tecach K-frames. We use a lightweight MLLM (Qwen2.5-VL-3B) as the backbone. The initial Supervised Fine-Tuning (SFT) stage prepares the model for our scene-driven paradigm by instilling foundational capabilities in temporal localization and scene understanding. Then during the second SFT stage the model learns with supervised data to perceive query-relevant video clips with reason, enabling our clip2frame prediction. Finally, the SFT-trained model serves as a cold-start policy for Reinforcement Learning, where the scene-driven keyframe selection policy is directly optimized to ensure the selected scenes are maximally effective for downstream task. This entire process yields a model that outputs query-conditioned key clips rather than disconnected frames, naturally enabling interpretable and flexible any-k keyframe selection.

To sum up, the main contributions are: (1) We construct PeakClips, a 200K query-conditioned highlight dataset built via scene segmentation, hierarchical captioning, and LLM-guided relevance scoring, providing supervision for temporal grounding, scene perception, and keyframe prediction. (2) We propose K-frames, a new interpretable paradigm that reframes keyframe selection as clip2frame prediction, preserving scene continuity and enabling any-k keyframe selection. (3) Extensive experiments on major long-video understanding benchmarks demonstrate that K-frames provides an effective, interpretable, and plug-and-play solution for keyframe selection at multi-scales.

## 2 RELATED WORK

### 2.1 MULTI-MODAL LARGE LANGUAGE MODELS FOR VIDEO UNDERSTANDING

Existing MLLMs such as ChatGPT-4o Hurst et al. (2024), Gemini 2.5 Pro Comanici et al. (2025) and Qwen-VL 2.5 Bai et al. (2025) have made significant progress in multimodal understanding (Achiam et al., 2023; Team et al., 2023; Bai et al., 2023). However, adapting these models to the video domain

introduces the added complexity of modeling temporal information. Early efforts in video-MLLMs primarily relied on uniformly sampled frames and simple connectors, such as MLPs (Lin et al., 2023; Ataallah et al., 2024; Maaz et al., 2024b), discrete visual tokenizers (Jin et al., 2024) and Q-formers (Zhang et al., 2023; Li et al., 2024b) to link visual encoders with LLMs. Subsequent models focus on enhanced video-instruction data (Li et al., 2024a; Wang et al., 2024), efficient spatio-temporal feature compression methods (Shen et al., 2024; Tan et al., 2024) and video-specific encoders (Wang et al., 2024; Maaz et al., 2024a). Specifically, processing long videos remains a significant bottleneck due to MLLMs' context limits and prohibitive computational costs. Current strategies to mitigate this challenge include directly extending the LLM's context window (Zhang et al., 2024a), developing memory management mechanisms (He et al., 2024) or keyframe selection algorithms (Tang et al., 2025; Lee et al., 2025; Xu et al., 2025) for identifying representative frames.

## 2.2 EXISTING KEYFRAME SELECTION METHODS

Efficient keyframe selection has become a critical component for long-video understanding, evolving from traditional approaches like query-agnostic clustering-based methods (Zhang et al., 2013) or uniform sampling (Xu et al., 2024) to modern query-adaptive strategies. They are predominantly divided into two paradigms: text-image retrieval and RL-based frame optimization. Text-image retrieval methods calculate the independent video frame-query similarity to localize important frames. MLLM Based Frame Selection (Hu et al., 2025) employs spatial-temporal importance scoring to boost performance, and Frame-Voyager (Yu et al., 2024) ranks frame combinations via pretrained Video-LLMs. Concurrently, there have been endeavors to integrated RL into keyframe selection for policy optimization. ReFoCUS (Lee et al., 2025) proposed a frame-level policy optimization framework that shifts the optimization target from textual responses to visual input selection, and ViaRL (Xu et al., 2025) leverages the downstream model's answer accuracy as a reward signal, enabling a trial-and-error learning that requires no explicit frame selection annotations. Yet, these approaches prioritize frame-level semantics, largely ignoring a video's crucial temporal structure. In contrast, our method K-frames redefines this task through clip2frame prediction, a paradigm that preserves the narrative flow of events and supports versatile any-k selection.

## 3 METHOD

In this work, we propose K-frames, which reframes keyframe selection as the task of predicting query-relevant key clips and sampling frames. To achieve this, our model needs to understand scene-level semantics and their temporal boundaries. A main challenge, however, is the lack of datasets with scene-level relevance annotations. To address this, we first present the construction of our large-scale dataset, PeakClips, which provides the necessary supervision (Sec. 3.1). Building on this dataset, we train K-frames using a novel three-stage progressive curriculum. We begin with two stages of Supervised Fine-Tuning to equip the model with the fundamental capabilities of temporal grounding and key-clip perception (Sec. 3.2). Finally, we employ Reinforcement Learning to align the model's clip2frame selection policy with downstream long-video understanding tasks, without the need for further annotations (Sec. 3.3). The overall system is illustrated in Figure 3.

## 3.1 PEAKCLIPS DATASET

To enable our clip-to-frame learning paradigm, we introduce **PeakClips**, a large-scale dataset comprising over 200K query-conditioned relevance annotations on video clips. The source videos are drawn from LLaVA-Video-178K (Zhang et al., 2024b), NeXT-QA (Xiao et al., 2021), and PerceptionTest (Patraucean et al., 2023). As illustrated in Figure 2, we follow a three-stage pipeline: given a video $v$ and a text query $q$, we first segment $v$ into candidate clips, then estimate the relevance of each clip to $q$, and finally retain only the most relevant ones. This yields, for each video $v$, a set of $N_v$ key clips,

$$\mathcal{C}_v = \big\{\, c_v^{(i)} = [s_v^{(i)}, e_v^{(i)}] \,\big\}_{i=1}^{N_v},$$

where the superscript $i$ indexes the key clips within video $v$, $s_v^{(i)}$ and $e_v^{(i)}$ denote the start and end frame indices of the $i$-th key clip, and $N_v$ is the number of selected key clips for video $v$.

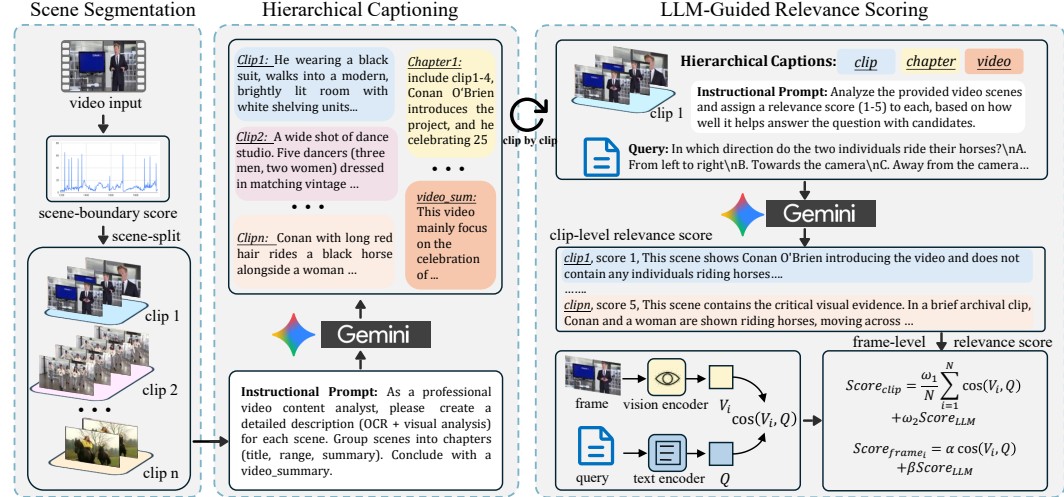

Figure 2: The three-stage framework for constructing the PeakClips dataset. The process involves (1) Scene-aware Segmentation to partition the video, (2) Hierarchical Captioning to generate multi-level descriptions, and (3) LLM-guided Relevance Scoring to identify query-conditioned relevance.

**Scene-aware Segmentation.** We first decompose each video into a set of temporally contiguous and semantically coherent scenes. To achieve this, we calculate the change in visual content throughout the video by computing the histogram difference between consecutive frames (Sheena & Narayanan, 2015). This process generates a scene-boundary score for each frame transition, where high scores correspond to abrupt changes in visual content. By segmenting the video at these high-scoring boundaries $\{b_v^0 = 1, b_v^1, \ldots, b_v^M\}$, we obtain a set of scene clips $s_v^j = [b_v^{j-1}, b_v^j]$.

**Hierarchical Captioning.** To provide multiscale context for relevance scoring, we generate captions through Gemini 2.5 Pro (Comanici et al., 2025) at three granularities: **fine-grained clip-level descriptions**, **chapter-level summaries** (grouping related clips), and a **video-level synopsis**. This hierarchy allows relevance to be assessed by connecting local events to the global narrative, which is crucial for handling complex queries.

**LLM-guided Relevance Scoring.** With the segmented clips and captions, we first use Gemini 2.5 Pro to assign a base relevance score (1-5) with a reason to each clip based on a detailed instructional prompt (see Appendix C.2). This LLM-generated score is then refined using the text-frame similarity. Specifically, we compute a final clip-level score by taking a weighted average of the LLM score and the mean SIGLIP (Zhai et al., 2023) similarity between the query and the clip's frames. We are able to extend the relevance score to frame level by weighting the parent clip's Gemini score with each frame's individual SIGLIP-query similarity. But in our work, we only use the clip-level relevance. Clips with a final score greater than or equal to 4.9 are annotated as top-priority (**P1**) highlights, while those with scores in the range $[4.3, 4.9)$ are labeled as secondary-priority (**P2**) clips.

**PeakClips Dataset.** In summary, the three-stage construction pipeline yields the **PeakClips** dataset, a comprehensive resource for video understanding. Each entry provides videos annotated with temporally coherent scene boundaries, multi-level hierarchical captions (clip, chapter, and video), and query-conditioned highlight clips. The dataset also includes the dense, continuous clip-level relevance scores and LLM-generated rationales that informed the final selections. Collectively, these rich annotations make PeakClips a versatile resource for supervising a wide spectrum of tasks, including temporal grounding, scene-level perception, keyframe selection.

### 3.2 SUPERVISED FINE-TUNING FOR KEY-CLIP PREDICTION

Building on the PeakClips dataset, K-frames learns scene-driven keyframe selection through a three-stage progressive curriculum (Bengio et al., 2009). As illustrated in Figure 3, it includes two-stage

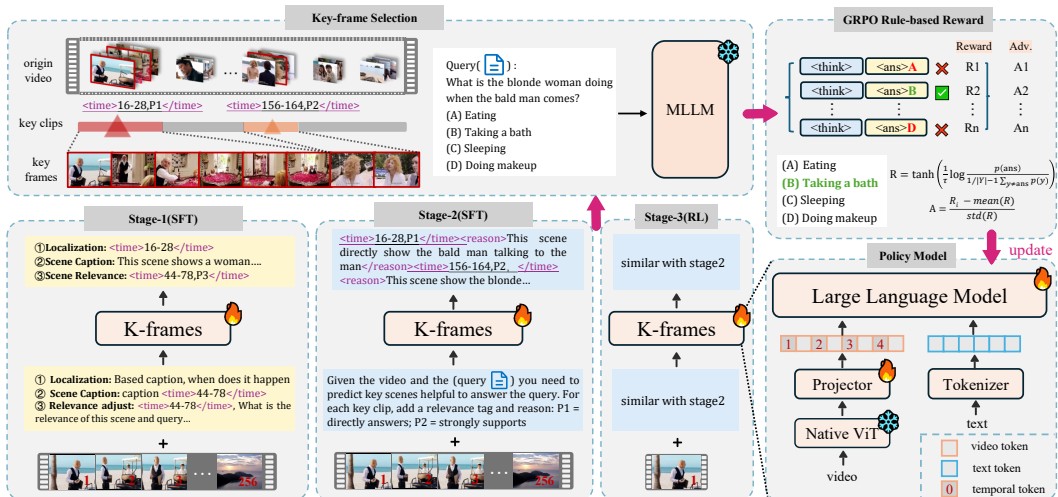

Figure 3: An overview of the K-frames framework. It features a two-stage Supervised Fine-Tuning (SFT) curriculum for temporal grounding and key-clip perception, followed by a Reinforcement Learning (RL) stage to align the selection policy with downstream task performance.

Supervised Fine-Tuning (SFT) and one-stage Reinforcement Learning (RL). The first two stages is to equip a lightweight MLLM(Qwen2.5-VL-3B) with two core capabilities essential for our task: temporal localization, and query-conditioned key clips perception.

**Temporal Grounding and Relevance Judge.** In the first SFT stage, K-frames leverage the hierarchical captions and clip-query relevance annotations in PeakClips to learn Temporal Grounding. To enhance the K-frames' ability to align visual content with its time span, we employ two temporal prompting techniques throughout all three stages training. Following prior work (Wu et al., 2025), Visual Prompts render the frame index $t$ directly onto each frame $f_t$, providing a direct visual cue for time. Concurrently, we inject Textual Prompts preceding the visual tokens $\mathbf{v}_{t,t+1}$ for each frame. Building on these temporal cues, our curriculum is designed to instill robust localization and perception abilities.

To directly enhance the model's temporal localization capabilities, we design a **caption-to-scene localization** task, where the model receives the video and a scene description to locate its temporal span. As a dual task, we introduce a **scene-to-caption generation**, requiring the model to generate a description for a given temporal span. Moreover, we incorporate a **clip-query relevance scoring** task. In this task, the model is required to predict how relevant a specific clip from the whole video is to a given query. The full specific instruction prompt for these three tasks see Appendix D.1.

**Query-Conditioned Key-Clip Prediction.** Building upon the foundational abilities learned in stage 1, the second SFT stage teach our model with parameters $\theta$ for its ultimate goal: given a long video $V = \{f_t\}_{t=1}^{T}$ and a query $Q$, it learns to perceive and predict a set of relevant key clips $\mathcal{C} = \{c_i\}_{i=1}^{N}$. Each predicted clip $c_i$ consists of a temporal span and a textual rationale. In this training phase, the model is conditioned on the full video $V$, the query $Q$, and a specific instruction prompt $I$. The prompt instructs the model to select query-relevant video clips, assigning a priority tag (**P1** for direct answers, **P2** for strong support), and providing a brief rationale for each selection (see Appendix D.1 for the full prompt text).

The training in both SFT stages is unified under a standard auto-regressive language modeling objective. The model is optimized to maximize the likelihood of generating the ground-truth sequence $\mathcal{Y}_{\text{gt}}$ (Mao et al., 2023):

$$\mathcal{L}_{\text{SFT}} = -\log P(\mathcal{Y}_{\text{gt}}|V, Q, I; \theta) \tag{1}$$

This holistic training compels the model to predict key clips conditioned on query. This process yields a well-initialized policy for the subsequent Reinforcement Learning stage and provides a strong, standalone model for key clip selection.

## 3.3 REINFORCEMENT LEARNING FOR DOWNSTREAM TASK ALIGNMENT

To bridge the gap between mimicking annotations from Supervised Fine-Tuning and maximizing downstream task performance, we introduce a Reinforcement Learning stage. This stage directly optimizes the K-frames policy by aligning it with the final task objective, using the SFT-trained model as the initial policy. This alignment process requires no further annotations.

**Scene-driven Keyframe Selection.** The RL process begins with our SFT-trained K-frames, which functions as the actor model. For a given video $V$ and query $Q$, the actor model predicts a set of key clips. From these predicted clips, which represent the most informative segments, we then sample a fixed budget of $k$ keyframes using uniform sampling. This clip-first, sample-second strategy ensures that the selected frames are both semantically relevant and temporally coherent. These $k$ keyframes, along with the original query, are then fed into a powerful, frozen downstream MLLM (Qwen2.5-VL-7B) to generate a final answer to the query. The goal of our RL curriculum is to optimize the actor's clip2frame selection policy to maximize the quality of this final answer.

**Policy Optimization with GRPO.** To optimize our scene-driven keyframe selection policy, we employ Group Relative Policy Optimization (GRPO) (Shao et al., 2024), which eliminates the need for an explicit critic model by rolling out multiple candidate key clip selections and estimating their relative advantages. Instead of relying on a separate reward model, we compute a reward signal directly from the downstream model's output using a rule-based reward function. We perform this RL optimization exclusively on multiple-choice question-answering datasets to ensure a stable and reliable reward signal. Given a query $q$ and $G$ groups of rollout outputs $\{o_1, ..., o_G\}$ by our keyframe selector. The reward $\mathcal{R}$ evaluates answer quality by comparing the log-probability of the correct token against the average log-probability of incorrect ones, smoothed via a $\tanh$ transformation and a temperature hyperparameter $\tau$:

$$\mathcal{R}(q, o_i) = \tanh\left(\frac{1}{\tau}\log\frac{p_{t=\hat{y}}}{\frac{1}{|Y|-1}\sum p_{t\neq\hat{y}}}\right) \tag{2}$$

where $|Y|$ is the size of the candidate answer set and the probabilities $p(\cdot)$ are from the frozen downstream MLLM. We adopt the Dr. GRPO (Liu et al., 2025) variant to improve training stability and shorten the reasoning length. The group-relative advantage $A_i$ for each rollout defined in GRPO is calculated by:

$$\hat{A}_{i,t} = R(q, o_i) - \text{mean}\left(\{R(q, o_1), \ldots, R(q, o_G)\}\right) \tag{3}$$

Overall, the training objective for the RL stage is:

$$\mathcal{L}_{\text{RL}} = \frac{1}{G}\sum_{i=1}^{G}\sum_{t=1}^{|o_i|}\left\{\min\left[\frac{\pi_\theta(o_{i,t} \mid q, o_{i,<t})}{\pi_{\theta_{\text{ref}}}(o_{i,t} \mid q, o_{i,<t})}\hat{A}_{i,t}, \ \text{clip}\left(\frac{\pi_\theta(o_{i,t} \mid q, o_{i,<t})}{\pi_{\theta_{\text{ref}}}(o_{i,t} \mid q, o_{i,<t})}, 1-\epsilon, 1+\epsilon\right)\hat{A}_{i,t}\right]\right\}, \tag{4}$$

where $\pi_{\theta_{\text{ref}}}$ and $\pi_\theta$ denotes the reference model and the actor model in the GRPO framework.

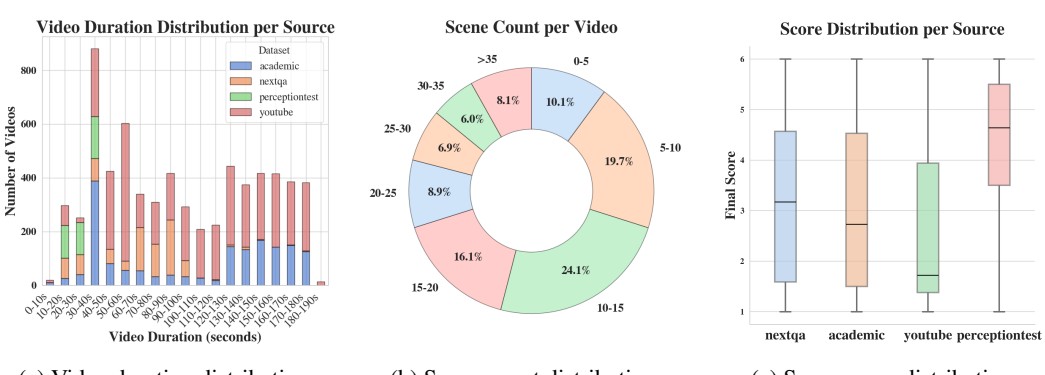

(a) Video duration distribution.  (b) Scene count distribution.  (c) Scene score distribution.

Figure 4: Statistics about our proposed PeakClips dataset.

**Any-K Keyframe Sampling.** The full training pipeline yields the final, optimized K-frames model, which serves as a versatile, plug-and-play model for long-video understanding at inference. The process begins with K-frames predicting query-relevant key clips. Based on these predictions, we support two flexible keyframe sampling strategies: Focused Sampling (exclusively from key clips) and Hybrid Sampling (densely from key clips, sparsely from the rest of the video). This provides a flexible trade-off between deep focus and broad context. More detailes see in Appendix D.2. These two strategies offer a flexible trade-off between concentrating on critical moments and maintaining broader video context.

## 4 EXPERIMENTS

### 4.1 PEAKCLIPS STATISTICS

The **PeakClips** dataset consists of more than 200k annotations, derived from 6,702 randomly selected videos from the LLAVA-Video-178K (Zhang et al., 2025) and labeled by Gemini 2.5 Pro. Specifically, our dataset consists of 108,221 scenes with 281,643 corresponding scene-query relevance annotations, and 19,070 chapters, with an average of 16.15 scenes and 2.85 chapters per video. As shown in Figure 4a, video durations vary notably across sources: PerceptionTest and NextQA clips are generally short, typically below 100 seconds, while Academic and YouTube videos exhibit a wider range with both short and long instances represented. Figure 4b illustrates the scene count distribution per video: the majority fall within 5–15 scenes (43.8%), while 10.1% contain fewer than 5 scenes and 8.1% exceed 35 scenes, indicating a balanced decomposition into semantically meaningful units. Finally, Figure 4c reports the scene–query relevance scores produced by the LLM. PerceptionTest videos achieve the highest consistency with a median above 4.5, YouTube clips show broader variance with lower medians, and Academic and NextQA datasets lie between these extremes. For more dataset statistics please refer to Appendix C.2.

### 4.2 EXPERIMENT SETUP

**Evaluation Benchmarks.** We conduct experiments on three public benchmarks to evaluate our approach. Video-MME (Fu et al., 2025) comprises 900 videos and 2,700 multiple-choice Question-Answer pairs, categorized into three subsets based on video duration: short ($<2$ minutes), medium (4-15 minutes), and long (30-60 minutes). MLVU (Zhou et al., 2025) includes videos ranging from 3 minutes to 2 hours and spans 9 tasks, with 2,174 multiple-choice VQA pairs. LongVideoBench (Wu et al., 2024) features videos with an average duration of 4,101 seconds per video, which is the longest. It contains 1,549 multiple-choice VQA pairs across 6 tasks. Importantly, all datasets are human-annotated, ensuring high-quality labels for evaluation. To verify model-agnostic generality, we evaluated downstream tasks with open-source models, including Qwen2.5-VL-7B, Qwen2.5-VL-72B (Bai et al., 2025) and Intern3.5-VL-8B (Wang et al., 2025); and closed-source models comprise ChatGPT-4o and Gemini 2.5 Pro (Comanici et al., 2025).

**Implementation Details.** We train K-frames with Qwen2.5-VL-3B as the backbone. For each training and evaluation instance, we uniformly sample $T = 256$ frames per video as inputs. K-frames predict continuous index ranges $[s, e]$ as highlight clips together with rationales. These key clips then guide the selection of $k$ keyframes for the downstream task. Specifically, when the number of keyframes set to $k = 8$, we employ Focused Sampling (exclusively from key clips). When $k = 32/64$, we employ Hybrid Sampling (details in Appendix 12). As described in Sec. 3.1, we construct PeakClips for our Supervised Fine-Tuning, and randomly select 20K samples from original LLaVA-Video-178K for our Reinforcement Learning optimization. During all 3 stages training, we freeze the vision encoder and update only the multimodal projector and the LLM. The first two supervised phase takes 36 hours, and the RL phase 40 hours. We use a learning rate of $1.0 \times 10^{-5}$ for both supervised phases, and $1.0 \times 10^{-6}$ for RL with a KL penalty coefficient of 0.01.

### 4.3 PERFORMANCE ACROSS GENERAL VIDEO BENCHMARKS

**Temporal Localization.** We first evaluate our K-frames on the Needle QA (a subset of MLVU). It constructs each example by randomly inserting a short "needle" segment containing evidence into a longer background video and annotating a corresponding pair of questions and answers, thereby directly probing temporal grounding. As shown in Table 1, compared to uniform sampling, our

Table 1: Main results on long-video understanding benchmarks. Our method (K-frames) consistently improves the performance of various open-source and closed-source MLLMs across different frames. The red text indicates the performance improvement over the baseline (uniform smpling). And the purple background highlights the largest improvement over the baseline.

| Models | Size | Frames | MLVU | | VideoMME | | | | LVBench |
|---|---|---|---|---|---|---|---|---|---|
| | | | Needle-QA | M-Avg | Short | Medium | Long | Avg | |
| VideoChat2 | 7B | 16 | - | 44.5 | 48.3 | 37.0 | 33.2 | 39.5 | - |
| VideoLLaVA | 7B | 8 | - | 47.3 | 45.3 | 38.0 | 36.2 | 39.9 | - |
| Video-CCAM | 14B | 96 | 73.2 | 63.1 | 62.2 | 50.6 | 46.7 | 53.2 | - |
| Video-XL | 7B | 128 | 73.8 | 64.9 | 64.0 | 53.2 | 49.2 | 55.5 | - |
| Kangaroo | 7B | 64 | - | - | 66.1 | 55.3 | 46.7 | 56.0 | - |
| VideoTree | - | - | - | 60.4 | 67.8 | 59.9 | 54.2 | 60.6 | - |
| ViaRL | 3+7B | 8 8 | 73.5 | 58.2 | 65.1 | 56.1 | 50.8 | 57.3 | - |
| open-sourced model | | | | | | | | | |
| Qwen2.5-VL | 7B | 8 | 58.6 | 53.9 | 61.7 | 50.6 | 46.9 | 53.0 | 52.8 |
| + languagebind | 7B | 8 | 51.6 | 52.3 | 54.3 | 49.2 | 45.9 | 49.8 | 52.2 |
| + ours | 3+7B | 8 | 77.5 (↑18.9) | 60.4 (↑6.5) | 68.9 | 55.3 | 47.9 | 57.4 (↑4.4) | 57.7 (↑4.9) |
| Qwen2.5-VL | 7B | 32 | 63.4 | 61.7 | 71.8 | 60.8 | 50.1 | 60.2 | 59.3 |
| + languagebind | 7B | 32 | 79.0 | 64.0 | 61.7 | 55.2 | 49.0 | 55.3 | 55.2 |
| + ours | 3+7B | 32 | 79.4 (↑16.0) | 65.9 (↑4.2) | 74.1 | 61.4 | 51.7 | 62.1 (↑1.9) | 60.5 (↑1.2) |
| Qwen2.5-VL | 7B | 64 | 67.7 | 65.6 | 73.9 | 62.3 | 52.2 | 62.8 | 59.9 |
| + ours | 3+7B | 64 | 78.9 (↑11.2) | 67.8 (↑2.2) | 75.9 | 63.7 | 53.9 | 64.5 (↑1.7) | 61.6 (↑1.7) |
| Qwen2.5-VL | 72B | 8 | 51.6 | 56.3 | 65.6 | 56.6 | 51.1 | 57.7 | 55.6 |
| + ours | 3+72B | 8 | 77.2 (↑25.6) | 63.3 (↑7.0) | 70.2 | 58.9 | 52.8 | 60.6 (↑2.9) | 59.3 (↑3.7) |
| Qwen2.5-VL | 72B | 32 | 67.3 | 64.0 | 74.3 | 63.4 | 58.1 | 65.3 | 60.8 |
| + ours | 3+72B | 32 | 78.3 (↑11.0) | 67.6 (↑3.6) | 75.2 | 66.0 | 57.8 | 66.3 (↑1.0) | 63.2 (↑2.4) |
| close-sourced model | | | | | | | | | |
| Gemini2.5Pro | – | 8 | 43.4 | 54.2 | 77.7 | 67.4 | 62.1 | 69.1 | 57.8 |
| + ours | – | 8 | 71.6 (↑28.2) | 56.6 (↑2.4) | 79.7 | 67.2 | 62.8 | 70.0 (↑0.9) | 62.2 (↑4.4) |
| Gemini2.5Pro | – | 32 | 74.6 | 66.0 | 87.1 | 74.9 | 69.6 | 77.2 | 64.2 |
| + ours | – | 32 | 80.9 (↑6.3) | 69.0 (↑3.0) | 87.1 | 76.1 | 70.9 | 78.0 (↑0.8) | 67.0 (↑2.8) |
| GPT-4o | – | 8 | 58.3 | 55.38 | 67.2 | 58.6 | 53.5 | 59.7 | 49.4 |
| + ours | – | 8 | 75.2 (↑16.9) | 60.5 (↑5.1) | 72.4 | 60.8 | 54.6 | 62.6 (↑2.9) | 54.5 (↑5.1) |
| GPT-4o | – | 32 | 71.3 | 59.6 | 69.3 | 61.1 | 55.8 | 62.1 | 49.9 |
| + ours | – | 32 | 76.9 (↑5.6) | 61.9 (↑2.3) | 70.6 | 62.7 | 54.8 | 62.7 (↑0.6) | 51.3 (↑1.4) |

K-frames significantly improves performance on this task. For example, when using the Gemini 2.5 Pro as the downstream model with the number of frames set to $k = 8$, our method boosts the accuracy from **43.4%** to **71.6%**, achieving a notable improvement of **28.2%**. This is because our model can effectively align visual evidence to time span and then locate the relevant scenes.

**Quantitative Analysis.** We evaluate K-frames on several challenging long-video benchmarks. As shown in Table 1, our method consistently and significantly exceeds the baseline performance in different open-source and closed-source models. For example, when applied to the open source QwenVL-2.5-7B with $k = 8$ frames, our approach achieves a improvement of **6.5%** on MLVU (M-Avg). Similarly, when integrated with the closed-source GPT-4o with $k = 8$, it boosts the LVBench score by a significant **5.1%**. This is because our model can effectively localize relevant scenes by aligning visual evidence with its correct time span, enabling it to effectively extract keyframes.

Furthermore, our method demonstrates robust performance gains even as the number of sampled keyframes increases. Taking the Qwen2.5-VL-72B's performance on LVBench as an example, the baseline score scales from 55.6 with 8 frames to 59.9 with 64 frames. Our method also improves upon these scores at each step—achieving 59.3 (**+3.7**) and 61.1 (**+1.2**) respectively. This demonstrates the scalability and effectiveness of our K-frames across different sampling densities.

Figure 5: Qualitative comparison between uniform sampling and our K-frames method.

Using the same QwenVL2.5-3B backbone for frame selection, we compare our method against ViaRL (Xu et al., 2025). It requires an iterative update strategy that involves jointly optimizing the downstream QwenVL2.5-7B model. In contrast, our method is truly plug-and-play, eliminating the need for costly downstream model optimization. As illustrated in Table 1, our approach outperforms ViaRL by **4.0** points on Needle-QA and **2.2** points on the MLVU M-Avg score. Moreover, unlike ViaRL's fixed 8-frame selection, our method can select any-k keyframes, which showcases the enhanced flexibility and superiority of our clip2frame paradigm. Unlike VideoTree—a training-free agent that relies on ChatGPT to caption and select video clips, resulting in high inference costs—our K-frames method adopts a lightweight QwenVL2.5-3B backbone for frame selection and can be plugged directly into any downstream MLLM. When integrated with GPT-4o, our approach surpasses VideoTree by **2.1** points on VideoMME.

**Qualitative Analysis.** Figure 5 presents a qualitative analysis that visually contrasts the performance of K-frames against the widely used uniform sampling baseline. For instance, when asked to identify the car's color in the video, uniform sampling method is blind to the query, selecting frames from various irrelevant scenes. In contrast, K-frames showcases a more sophisticated understanding. Its relevance score identifies two distinct but semantically related scene clips: one showing the "black funeral car driving" and another showing a "man in suit walking next to the black car", which provides the downstream model with comprehensive and unambiguous visual evidence. These examples demonstrate how K-frames successfully identifies and leverages critical visual evidence and leads to more accurate and well-grounded video understanding.

## 4.4 ABLATION STUDY

Table 2: Ablation on training stages. Baseline is uniformly sampling. Downstream MLLM is Qwen2.5-VL-7B with $k = 32$ frames.

| MODEL | SFT 1 | SFT 2 | RL | Needle-QA | MLVU |
|---|---|---|---|---|---|
| baseline | - | - | - | 63.4 | 61.7 |
| SFT | - | ✓ | - | 75.8 | 64.1 |
| SFT | ✓ | ✓ | - | 76.3 | 64.5 |
| RL | ✓ | ✓ | ✓ | **79.4** | **65.9** |

Table 3: Ablation on temporal prompts, performed on SFT2 model for efficient validation. The baseline uses a uniform sampling strategy.

| MODEL | TP | VP | Needle-QA | MLVU |
|---|---|---|---|---|
| baseline | - | - | 63.4 | 61.7 |
| SFT2 | ✓ | - | 75.5 | 63.9 |
| SFT2 | - | ✓ | 70.4 | 62.2 |
| SFT2 | ✓ | ✓ | **75.8** | **64.1** |

**Ablation on Training Stages.** We first analyze the contribution of each stage in our multi-stage progressive curriculum. As shown in Table 2, training with only the second SFT stage yields a score of 64.1 on MLVU. SFT2 is a necessary course because the model learns to predict query-conditioned key clips during SFT2, making it a prerequisite for the keyframe selection. SFT1 is a preliminary curriculum focused on foundational skills. Incorporating SFT1 provides a further gain, reaching 64.5 on MLVU and 76.3 on Needle-QA, which demonstrates this phase enhanced K-frames temporal grounding, which helps to final secene-driven keyframe selection. Moreover, adding the final Reinforcement Learning (RL) stage achieves a significant improvement, boosting performance by **1.4**% on MLVU and **3.1**% on Needle-QA. This is because RL stage directly optimizes its clip2frame selection policy to align with the downstream tasks.

Table 4: Main Inference-time on EgoSchema and MLVU. The (*) indicates an estimated time.

| Method | Dataset | Length (s) | Acc. | Inf. Time (s) |
|---|---|---|---|---|
| LangRepo | EgoSchema | 180 | 60.8 | 87.2 |
| VideoTree (Mistral-7B) | EgoSchema | 180 | 63.0 | 24.3 |
| VideoTree (Mistral-8×7B) | EgoSchema | 180 | 71.0 | 50.3 |
| K-frames | EgoSchema | 180 | – | 12.8 |
| VideoTree* (Mistral-7B) | MLVU | 930 | – | >24.3 |
| LanguageBind | MLVU | 930 | 52.3 | 11.2 |
| K-frames | MLVU | 930 | 60.4 | 10.6 |

**Ablation on Different Temporal Prompts.**   Given the importance of temporal cues, we next explore the individual contributions of our two temporal prompts: Visual Prompt (VP) and Textual Prompt (TP). As shown in Table 3, using only VP or TP results in suboptimal performance. This limitation suggests that relying on a single type of prompt provides an incomplete representation of the video's temporal structure. In contrast, combining both VP and TP attains a final score of **64.1**% on MLVU. It is because our two prompts capture complementary information. The VP directly provides visual evidence, while the TP offers fine-grained, position-specific guidance for each frame. This synergy allows K-frames to build a more comprehensive understanding of temporal dynamics, enhancing its scene-driven keyframe selection performance.

## 4.5 INFERENCE-TIME ANALYSIS

Table 4 compares the accuracy-latency trade-off of K-frames against other keyframe selecting methods. On MLVU, K-frames raises accuracy from **52.3** to **60.4** compared to LanguageBind, while slightly reducing inference time from 11.2s to 10.6s—demonstrating that our scene-driven selector improves performance with Limited computational overhead. In contrast, VideoTree relies on captioning multiple candidate shots and repeated LLM queries, incurring substantially higher latency. On EgoSchema, K-frames requires only 12.8s per video, compared to 24.3s for VideoTree (Mistral-7B). For longer videos, such as those in MLVU (average 930s), VideoTree's captioning cost scales up, leading to even higher estimated runtimes (denoted as "VideoTree*" in Table 4). K-frames, by contrast, maintains stable inference time due to its lightweight clip-to-frame selector.

## 5 LIMITATION

While K-frames significantly enhances long-video understanding, it still faces certain limitations. First, the current selector relies on Qwen2.5-VL-3B with a input budget of 256 frames, which may be too sparse for extremely long videos (e.g., over two hours), potentially causing important events to be undersampled. Scaling to such scenarios may require hierarchical or streaming mechanisms that process frames in multiple passes or incorporate long-term memory. Second, K-frames is scene-driven and works best for long videos with diverse scenes. For short clips with minimal scene changes, dense retrieval or exhaustive frame-combination strategies may be more effective. Future work could explore integrating K-frames with such complementary approaches.

## 6 CONCLUSION

In this work, we introduce K-frames, a new scene-driven paradigm for long-video understanding. It reframes keyframe selection as a clip2frame prediction task, preserving scene continuity while enabling flexible any-k sampling. To realize this paradigm, we first construct PeakClips, a new 200K query-clip relevance dataset. We then propose a three-stage SFT-RL training framework designed to produce a powerful key clip predictor that is highly optimized for downstream tasks. Extensive experiments show K-frames acts as an effective, interpretable, and model-agnostic front-end, consistently boosting MLLM performance on major long-video benchmarks.

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

## A   USE OF LARGE LANGUAGE MODELS (LLMS)

During the preparation of this manuscript, we utilized a Large Language Model (LLM) as a writing assistant. The primary application of the LLM was for language enhancement, which included improving grammar, refining wording for conciseness, and rephrasing sentences to improve clarity and flow.

In accordance with the established ethical guidelines, we, the authors, affirm that we are fully responsible for the content of this submission. All text, including any passages refined with the assistance of the LLM, has been critically reviewed, edited, and validated by the authors. The scientific claims, results, and conclusions presented herein are our own. We are solely responsible for any potential errors, inaccuracies, or ethical violations in this work.

## B   COMPARISON WITH PRIOR WORK

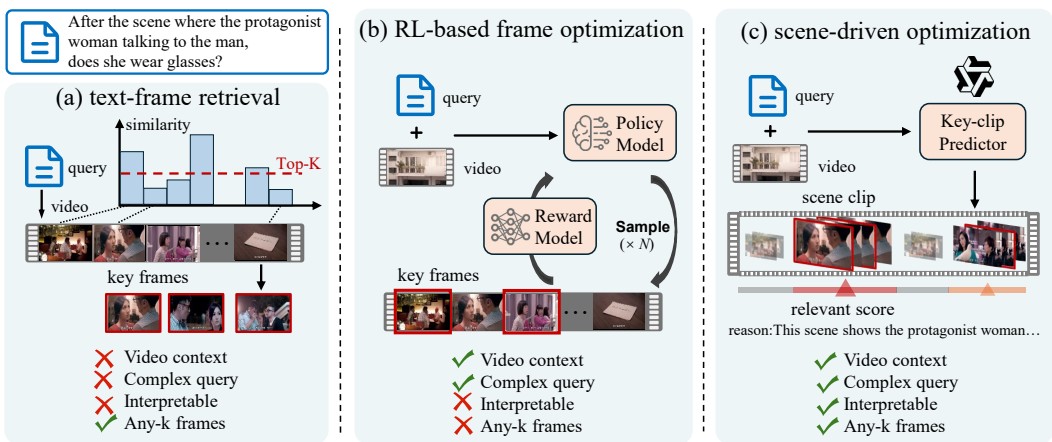

Figure 6: Comparison with existing keyframe selection methods.

As illustrated in Figure 6, our scene-driven K-frames paradigm addresses the main limitations of existing approaches. Text–frame retrieval methods treat videos as independent frame sets and rank them by query similarity, overlooking temporal context and offering limited interpretability. RL-based frame optimization considers selection as a combinatorial search that often yields a sparse, disconnected set of frames, breaking scene continuity and degrading downstream performance. They are also typically tuned for a fixed number of frames, lacking flexibility for any-k selection and making it difficult to meet personalized compute budgets. In contrast, our K-frames predicts semantically coherent, query-relevant clips and then samples keyframes, inherently preserving temporal continuity and providing interpretable clip-level rationales. Moreover, it supports flexible any-k selection, allowing users to balance performance and computational cost.

## C   DATASET CONSTRUCTION AND ANALYSIS

### C.1   IMPLEMENTATION DETAILS

In this section, we present how we organize our prompt to generate labels using LLM.

**Caption Generation.**   To obtain fine-grained scene-level descriptions after video segmentation, we employed an instruction-following style prompt, in which the model is explicitly assigned the role of a Professional Video Content Analyst. The prompt enforces a strict JSON output format containing three components: `scenes`, `chapters`, and `video_summary`.

As shown in Figure 7, our Instructional Prompt for caption generation is designed to guide the LLM through a structured, multi-stage analysis. The prompt instructs the model to first perform an initial skim for overall context, followed by a detailed scene-by-scene analysis that combines OCR of

on-screen text with a compositional description of visual elements . Subsequent instructions direct the model to refine scene boundaries by merging or splitting segments, aggregate related scenes into thematic chapters, and conclude with a high-level video summary . This step-wise, instruction-based format enforces a highly structured analytical process, resulting in objective and detailed video descriptions suitable for our dataset.

**Relevance Scoring.** To evaluate the relevance of each scene in the context of video question answering (VideoQA), we employed a second evaluation-oriented Instructional Prompt, positioning the model as a Video QA Relevance Analyst. The output is again required to follow a strict JSON structure, including the fields `scene_id`, `relevance_score`, and `reason`.

As illustrated in Figure 8, the procedure begins by providing the model with the question and the corresponding gold-standard answer, which serve as the reference criteria. Each scene is then assessed with respect to its contribution toward answering the question. Relevance is assigned according to a five-point ordinal scale:

- **5 (Directly Relevant)**: the scene contains critical visual evidence that directly resolves the question;
- **4 (Highly Relevant)**: the scene provides strong supporting context, though it is not the single most essential frame;
- **3 (Moderately Relevant)**: the scene depicts related subjects or environments but lacks the decisive information;
- **2 (Slightly Relevant)**: the scene has only weak or indirect connection to the question;
- **1 (Not Relevant)**: the scene provides no information useful for answering the question.

Each score must be accompanied by a concise justification (`reason`), ensuring interpretability and consistency across all annotations. This prompt design enforces rigorous evaluation criteria, quantitative scoring, and machine-readable outputs that are suitable for large-scale automated processing.

## C.2 DATASET STATISTICS

In this section, we present additional statistical details of the **PeakClips** dataset. We randomly sampled 6702 videos from LLAVA-Video-178K and adopted Gemini for the labeling. As listed in Table 5, the PeakClips dataset comprises over 200k annotations in total, including 6,702 annotated videos, 108,221 scenes with 281,643 corresponding relevance scores, and 19,070 chapters, with an average of 16.15 scenes and 2.85 chapters per video. Since the PeakClips dataset is derived from four sources—NextQA, Academic, YouTube, and PerceptionTest—we present in Table 6 the number of videos, scenes, and relevance scores associated with each source.

Table 5: Annotation statistics of the PeakClips dataset.

| Annotation Type | Count | Average per Video |
|---|---|---|
| Video-level Summarization | 6,702 | 1 |
| Chapter-level Description | 19,070 | 2.85 |
| Scene-level Description | 108,221 | 16.15 |
| Relevance Query | 16,883 | 2.52 |
| Scene-level Relevance Scores | 281,643 | 42.02 |
| **Total Annotations** | 281,643 | |

Below are two sample entries from the **PeakClips** dataset, illustrating (i) scene/chapter/video-level annotations (Figure 9) and (ii) scene–query relevance annotations (Figure 10).

## D TRAINING DETAILS

### D.1 INSTRUCTIONAL PROMPTS

This section provides the detailed instructional prompts used during the different stages of our training curriculum for K-frames.

You are a professional Video Content Analyst. Your primary task is to meticulously analyze a video based on a provided list of scene timestamps and generate a structured, detailed summary in a strict JSON format.
**Critical Rule:** The final output **MUST** be a single, valid JSON text.
**Output Format:** Strictly adhere to the following JSON structure:
**Analysis Steps:** Follow these reasoning steps to generate the final JSON output:
**[Step 1]:** Initial Skim & Contextual Understanding
Perform a quick overview of the entire video content ({{video}}) to understand its overall theme, setting, and subject matter. This initial context will help in creating logical chapters later.
**[Step 2]:** Detailed Scene-by-Scene Analysis
Iterate through each scene object provided in the {{timestamp}} list. The number of scenes you describe **must exactly match** the number provided in the timestamp data. For each scene, you must perform the following actions to generate a single combined description:
**Perform OCR**: Analyze the visual content of the scene to detect and transcribe any on-screen text.
**Describe Events & Composition**: First, analyze the scene's composition by identifying the key visual elements within the frame. Describe their positions and spatial relationships to convey the structure of the image. Then write a highly detailed and objective description of what is happening in the scene, ensuring the timeframe corresponds exactly to the start_time and end_time provided.
**Combine for Output**: The results from both actions must be combined into the single description field. If text was detected, describe it at the beginning, followed by the visual description.
**[Step 3]:** Scene Refinement
Review the list of described scenes from Step 2 to refine the segmentation. Only modify the original scene timestamps if you are absolutely certain it is essential for narrative clarity.
Merge: Combine adjacent, visually identical scenes that form a single, continuous action.
Split: Divide a scene only at an abrupt, hard cut to a new location or action.
Finalization: If any change was made, set the "relocate" flag to "1" and re-number all scene_ids sequentially. Otherwise, "relocate" is "0".
**[Step 4]:** Thematic Chapter Aggregation
After analyzing all individual scenes, review your descriptions and group consecutive scenes into logical chapters. A "chapter" should represent a distinct phase of the narrative, a change in location, or a cohesive block of related actions. For each chapter, create a chapter object with the following fields:
id: A unique sequential identifier for the chapter.
title: A concise, thematic title that captures the chapter's main focus.
scene_range: A string representing the range of scene IDs included in this chapter (e.g., "1-4").
description: A summary of the chapter's theme, explaining the story or process it covers (e.g., "This chapter follows the protagonist's journey through the forest.").
**[Step 5]:** Final Video Summary
Based on your chapter summaries, compose a final, high-level video_summary. This summary should concisely explain the entire video's purpose and narrative from beginning to end.
**[Step 6]:** Final JSON Assembly
Assemble all the generated data into the single, valid JSON text specified in the Output Format. Ensure all keys, brackets, and commas are correct.

Figure 7: Prompt used for generating scene-level captions.

Table 6: Scene relevance score statistics of PeakClips dataset across sources.

| Source | Videos | Scenes | Score 1 | Score 2 | Score 3 | Score 4 | Score 5 | Total |
|---|---|---|---|---|---|---|---|---|
| **Global** | 6,702 | 281,643 | 88,588 | 67,842 | 28,154 | 36,220 | 36,263 | 281,643 |
| NextQA | 716 | 10,444 | 2,004 | 2,460 | 1,453 | 1,713 | 1,648 | 10,444 |
| Academic | 1,512 | 72,961 | 17,791 | 16,168 | 8,436 | 11,676 | 11,817 | 72,961 |
| YouTube | 4,086 | 193,976 | 68,594 | 48,833 | 17,816 | 21,903 | 21,624 | 193,976 |
| PerceptionTest | 388 | 4,262 | 199 | 381 | 449 | 928 | 1,174 | 4,262 |

**Instructional Prompts for SFT1.** In the first SFT stage, we employ three task-specific prompts to instill foundational temporal grounding capabilities in the model. Each prompt is designed to teach a core sub-task.

Caption-to-Scene Localization. This task trains the model to identify the temporal boundaries (start and end frames) of a scene given its textual description. The prompt used is:
Scene-to-Caption Generation. As a dual task, this prompt instructs the model to generate a concise and accurate description for a given temporal segment of the video. The prompt used is:
Clip-Query Relevance Scoring. This task requires the model to assess and score the relevance of a specific video clip in relation to a given query, helping it learn to weigh the importance of different scenes. The prompt used is:

**Instructional Prompts for SFT2 and RL.** The second SFT stage uses a comprehensive prompt to train the model for its primary goal: predicting a complete set of highlight clips for a given video

You are a Video QA Relevance Analyst. Your task is to analyze a video, which has been broken down into timed scenes, and evaluate how relevant each scene is to answering a specific question.

For each scene, you will assign a relevance score from 5 (most relevant) to 1 (least relevant). Your final output must be a single, valid JSON object.

**Critical Rule:** The scene_id in your output must exactly match the scene_id from the provided scene data.

**Output Format:** Strictly adhere to the following JSON structure:

{"relevance_analysis":[{"scene_id":1,"relevance_score":1,"reason":"A brief explanation."},
{"scene_id":2,"relevance_score":5,"reason":"This scene directly shows the action."}]}

video:{{video}}}

video description: {{video_description}}}

Analysis and Scoring Instructions

**[Step 1]:** Understand the Core Task

First, carefully review the user's question and the provided correct answer. Your goal is to identify which video scenes contain the visual evidence needed to arrive at that correct answer.

Question: {{question}}

Correct Answer: {{answer}}

**[Step 2]:** Analyze Each Scene Against the Question

You will now evaluate each scene one by one. Use the scene's description and timing to determine its relevance to the question and answer.

**[Step 3]:** Assign a Relevance Score

Use the following 5-point scale to score each scene. Be strict and objective in your evaluation.

Score 5 (Directly Relevant): The scene contains the most critical visual information that directly and unambiguously answers the question. Without this scene, the question would be difficult or impossible to answer.

Score 4 (Highly Relevant): The scene provides strong, supporting visual context that reinforces the correct answer, but it may not be the single most crucial scene. It clearly shows an important part of the activity in question.

Score 3 (Moderately Relevant): The scene shows the subject or environment related to the question but does not show the key action itself. It provides context but is not sufficient to answer the question on its own.

Score 2 (Slightly Relevant): The scene has a weak or indirect connection to the question. For example, it might show the person just before or after the main activity, or focus on a background element.

Score 1 (Not Relevant): The scene contains no visual information that helps answer the question. It might be an establishing shot, a shot of an empty room, or an unrelated action.

**[Step 4]:** Assemble the Final JSON Output

Compile the analysis for all scenes into the final JSON format specified above. Ensure every scene from the input data has a corresponding entry in your output.

Figure 8: Prompt used for generating scene-level query relevance scores.

```
{
  "video_duration": 15.9159,
  "scene_count": 4,
  "scenes": [
    {"scene_id": 1, "start_sec": 0.0, "end_sec": 3.971, "duration_sec": 3.971, "start_frame": 0, "end_frame": 119},
    {"scene_id": 2, "start_sec": 3.971, "end_sec": 7.941, "duration_sec": 3.971, "start_frame": 119, "end_frame": 238},
    {"scene_id": 3, "start_sec": 7.941, "end_sec": 11.945, "duration_sec": 4.004, "start_frame": 238, "end_frame": 358},
    {"scene_id": 4, "start_sec": 11.945, "end_sec": 15.916, "duration_sec": 3.971, "start_frame": 358, "end_frame": 477}],
  "annotation": {
    "scenes": [
      {"id": 1, "start_time": "0.0", "end_time": "3.971", "description": "A woman and a young child stand in a brightly lit playroom
with wood-like floors and walls decorated with colorful cartoon stickers. The woman, on the left, wears a purple long-sleeved shirt, a
dark puffer vest, a light blue knit scarf, and blue jeans. She is smiling and speaking, gesturing with her hands as if starting a song. The
child, on the right, wears an orange hooded sweatshirt and blue pants, standing with their hands clasped in front of them."},
      ...
      {"id": 4, "start_time": "11.945", "end_time": "15.916", "description": "The woman turns back to the front, laughing and raising
her hands with palms open in a joyful, expressive gesture. The child, seemingly excited by the song's climax, begins to stomp their feet
and move around in a small circle, looking down at their feet before glancing back at the woman. The scene ends with both of them in
the middle of their respective actions."}],
    "chapters": [
      {"id": 1, "title": "A New Year's Song and Dance", "scene_range": "1-4", "description": "This chapter captures a heartwarming
moment between a woman and a young child in a playroom. The woman leads the child in singing a festive song, likely about the New
Year, complete with expressive hand gestures and dance moves. The child enthusiastically attempts to follow along, creating a charming
and lively scene."}],
    "video_summary": "This short video features a woman, likely a teacher or mother, teaching a young child a celebratory song and
dance in a colorfully decorated playroom. The woman sings and performs various hand gestures, which the child observes and tries to
imitate with enthusiasm. The video showcases a joyful interaction focused on learning and celebration through music and movement."
  }
}
```

Figure 9: An example data of Scene/Chapter/Video-level annotation by LLM in PeakClips.

```
{
    "index": "3883_1",
    "id": "EoSZu0UqLSA",
    "conversations": [
        {"from": "human", "value": "Which direction does the man enter the room from?\nA. From the back\nB. From the front\nC. From the right\nD. From the left\nPlease provide your answer by stating the letter followed by the full option."},
        {"from": "gpt", "value": "D. From the left."}],
    "anno_scenes": [
        {"scene_id": 1, "start_sec": 0.0, "end_sec": 14.556, "final_score": 5.5},
        {"scene_id": 2, "start_sec": 14.556, "end_sec": 29.071, "final_score": 1.26},
        ... ,
        {"scene_id": 11, "start_sec": 145.395, "end_sec": 159.951, "final_score": 1.96}],
    "llm_score": {
        "relevance_analysis": [
            {"id": 1, "score": 5, "reason": "This scene provides the direct and unambiguous visual evidence to answer the question. At the very beginning of the scene (0:00-0:02), the man is seen walking into the frame from the left side of the screen and sitting down on the couch next to the woman."},
            {"id": 2, "score": 1, "reason": "This scene shows the man and woman already seated on the couch and arguing. The man's entrance has already occurred and is not shown."},
            ... ,
            {"id": 11, "score": 1, "reason": "This scene is an outro for a YouTube channel and shows a different woman against a white background. It is completely unrelated to the question."}]
        }
}
```

Figure 10: An example data of scene relevance score labeled by LLM in PeakClips.

<video>Given the following detailed account, identify the precise start and end frames in the video where these events occur. Your response must be in the format <time>frames start-end</time>. Description: '{scene['description']}

Figure 11: Instructional prompt for Caption-to-Scene Localization.

<video>Analyze the video segment from <time>frames {scene['start_idx']}-{scene['end_idx']}</time>. Provide a detailed, objective narrative of the events, describe the visual composition, and transcribe all visible on-screen text.

Figure 12: Instructional prompt for Scene-to-Caption Generation.

<video>You are a Video Analyst. For the question: "{question}", please evaluate the relevance of the video segment <time>frames {start_idx}-{end_idx}</time>.
Your task is to provide a relevance tag and a brief reason based on the following scale:
- P1 = The segment directly answers the question.
- P2 = The segment strongly supports answering the question.
- P3 = The segment provides useful context but not direct evidence.
- P4 = The segment is weakly or tangentially related.
- P5 = The segment is irrelevant.
Return your analysis in the format: <time>frames {start_idx}-{end_idx}, P<score_tag>,</time><reason>reason</reason>

Figure 13: Instructional prompt for Clip-Query Relevance Scoring.

and query. This same prompt is then used by the actor model during the Reinforcement Learning (RL) stage to generate actions (i.e., predict key clips).

The prompt instructs the model to identify all relevant clips, assign a priority level (**P1** or **P2**) to each, and provide a brief rationale for its selection. The prompt used is:

You are a Video Analyst. Given a video and a question, you need to segment the video into scenes and select the segments helpful to answer the question. For each segment, add a relevance tag and a brief reason:\n- P1 = directly answers\n- P2 = strongly supports\nReturn segments in chronological order as start_idx-end_idx, P1|P2, short reason.

Figure 14: Instructional prompt for Clip-Query Relevance Scoring.

## D.2 INFERENCE DETAILS

After predicting the initial set of query-relevant key clips, a subsequent step is required to select the final $k$ keyframes. This section details the two methodologies we propose for this task: **Focused Sampling**, which selects keyframes exclusively from the predicted clips, and **Hybrid Sampling**, which dynamically samples from both the predicted clips and the background regions.

**Focused Sampling.** Given a video frame sequence $\{f_1, \ldots, f_T\}$, K-frames first predicts a set of query-relevant key clips $\boldsymbol{c} = \{([a_j, b_j], t_j)\}_{j=1}^M$, where $t_j \in \{\mathrm{P1}, \mathrm{P2}\}$ denotes the importance type and $\ell_j = b_j - a_j + 1$ is the length. We select $k$ keyframes *exclusively* from these predicted clips. Let $(w_{\mathrm{P1}}, w_{\mathrm{P2}}) = (2, 1)$ be class weights (P1 is twice as important as P2). We allocate the per-clip budget by weighted proportion:

$$k_j = \mathrm{round}\left( K \cdot \frac{w(t_j)\, \ell_j}{\sum_{i=1}^M w(t_i)\, \ell_i} \right), \qquad \begin{cases} w_{\mathrm{P1}}, & t = \mathrm{P1}, \\ w_{\mathrm{P2}}, & t = \mathrm{P2}. \end{cases} \tag{5}$$

To prevent short P1 clips from receiving zero frames, we enforce a *P1-at-least-1* guarantee by borrowing from donors with $k_i > 1$ (prefer P2 donors); if the global budget is insufficient, the guarantee is relaxed. Inside each clip, we sample *Uniformly*: pick $k_j$ equally spaced frames from $\{f_{a_j}, \ldots, f_{b_j}\}$ under a chronological constraint (strictly increasing frame indices across clips). If the total selected frames are fewer than $k$ due to rounding or chronology constraints, we top up uniformly from the non-key region after the last picked index.

---

**Algorithm 1:** Focused Sampling

1 **Require** Predicted clips $\boldsymbol{c} = \{([a_j, b_j], t_j)\}_{j=1}^M$, target $k$, weights $(w_{\mathrm{P1}}, w_{\mathrm{P2}}) = (2, 1)$;
2 Merge adjacent same-type clips within tolerance $\tau = 2$ (reasons concatenated);
3 Compute $k_j$ by equation equation 5, fix rounding so that $\sum_j k_j = k$;
4 Enforce P1-at-least-1 by borrowing from donors with $k_i > 1$ (prefer P2 donors);
5 last_id $\leftarrow -\infty$;
6 **foreach** $j \leftarrow 1$ *to* $M$ **do**
7     $\mathcal{C}_j \leftarrow \{f_{a_j}, \ldots, f_{b_j}\}$ restricted to frame id $>$ last_id;
8     Select $k_j$ equally spaced frames from $\mathcal{C}_j$;
9     Update last_id to the largest picked id;
10 **if** *selected* $< K$ **then**
11     Top up uniformly from non-key frames with id $>$ last_id;
12 Return $k$ frames sorted by index;

---

Table 7: Focused Sampling hyperparameters.

| | |
|---|---|
| Merge tolerance $\tau$ | 2 |
| Segment weights | $w_{\mathrm{P1}} = 2,\ w_{\mathrm{P2}} = 1$ |
| P1 guarantee | at least one frame for P1 if budget allows |
| Chronology constraint | strictly increasing frame indices |

**Hybrid Sampling.** We partition all candidate frames $F$ into predicted frames $\boldsymbol{p}$ (inside key clips) and background frames $\boldsymbol{b}$ (the rest). We first allocate a global share between $\boldsymbol{p}$ and $\boldsymbol{b}$, then distribute the predicted share across clips as in Focused Sampling (uniform only). Let $a_{\mathrm{pred}}$ be the prediction-to-background length weight ($a_{\mathrm{pred}} : 1 = 4 : 1$ in our default) and let $r_{\min} \in [0, 1]$ be a lower bound on the predicted share (e.g., $r_{\min} = 0.5$). With $|\boldsymbol{p}|$ and $|\boldsymbol{b}|$ the available counts, we compute

$$k_{\boldsymbol{p}}^{\mathrm{raw}} = \mathrm{round}\left( K \cdot \frac{a_{\mathrm{pred}}\, |\boldsymbol{p}|}{a_{\mathrm{pred}}\, |\boldsymbol{p}| + |\boldsymbol{b}|} \right), \tag{6}$$

$$k_{\boldsymbol{p}} = \min\left( |\boldsymbol{p}|,\ \max\left( \lceil K r_{\min} \rceil,\ k_{\boldsymbol{p}}^{\mathrm{raw}} \right) \right), \tag{7}$$

$$k_{\boldsymbol{b}} = \min\left( |\boldsymbol{b}|,\ K - k_{\boldsymbol{p}} \right). \tag{8}$$

If $k_p + k_b < k$ due to upper caps, remaining slots are assigned to the side that still has capacity. Inside $p$, we further allocate $k_p$ across clips using equation 5 with the P1-at-least-1 guarantee, and select frames uniformly within each clip. From $b$, we select $k_b$ frames uniformly. The final set is deduplicated and sorted by frame index.

---

**Algorithm 2:** Hybrid Sampling

---

1 **Require** Predicted clips $s$, full candidates $F$, target $k$, weight $a_{\mathrm{pred}} = 4$, minimum ratio $r_{\min}$
   (e.g., 0.5);
2 Build a mask from $s$ to partition $F$ into $p$ and $b$;
3 Compute $k_p$ by equation 6–equation 7; set $k_b = k - k_p$ and cap by availability; top up if
   needed;
4 Distribute $k_p$ across clips via equation 5 with P1-at-least-1;
5 Uniformly select frames within each predicted clip to meet its allocation;
6 Uniformly select $k_b$ frames from $b$; union, deduplicate, sort;
7 **Return** $k$ frames.

---

Table 8: Hybrid Sampling hyperparameters and defaults.

| | |
|---|---|
| Pred:background weight $a_{\mathrm{pred}} : 1$ | $4 : 1$ |
| Minimum predicted ratio $r_{\min}$ | 0.5 (configurable) |
| Within-pred clip weights | P1:2, P2:1; P1-at-least-1 guarantee |
| Chronology constraint | strictly increasing frame indices |

# E   MORE EXPERIMENTAL RESULTS

In this section, we provide additional quantitative and qualitative experimental results on long-video understanding benchmarks to further validate the effectiveness and generalizability of our proposed keyframe selection method.

## E.1   MORE RESULTS ON LONG-VIDEO BENCHMARK

Table 9: More results on long-video understanding benchmarks. The red text indicates the performance improvement over the baseline (uniform sampling).

| Models | Size | Frames | MLVU | | VideoMME | | | | LVBench |
|---|---|---|---|---|---|---|---|---|---|
| | | | Needle-QA | M-Avg | Short | Medium | Long | Avg | |
| InternVL3.5 | 8B | 8 | 60.3 | 60.5 | 68.0 | 56.7 | 49.7 | 58.1 | 57.7 |
| **+ ours** | 8B | 8 | 72.7 (↑12.4) | 60.4 (↑6.5) | 71.4 | 59.0 | 50.7 | 60.4 (↑2.3) | 60.0 (↑2.3) |
| InternVL3.5 | 8B | 32 | 72.4 | 67.0 | 75.7 | 64.3 | 53.9 | 64.6 | 60.1 |
| **+ ours** | 8B | 32 | 74.9 (↑2.5) | 68.4 (↑1.4) | 75.9 | 64.2 | 55.1 | 65.1 (↑0.5) | 61.8 (↑1.7) |

As shown in Table 9, we further assess our method's generalizability by pairing it with InternVL-3.5. The consistent gains show that our scene-driven keyframe selection paradigm provides a provides an effective, interpretable, and plug-and-play solution for long video understanding.

As illustrated in Figure 15 and Figure 16, We further provide detailed visualizations of the results showing the sub-category performance on the MLVU and VideoMME datasets evaluated with the Qwen2.5-VL 7B model using 8 input frames. These results show that our model consistently improves performance across different task types on the evaluation benchmarks, with particularly notable gains on the Needle-QA localization task in MLVU. This result underscores the core strength of our approach: by predicting query-relevant clips, K-frames preserve the temporal continuity and focus on informative clips. Moreover, we observe no improvement in the Topic Reasoning task of MLVU and Information Synopsis task of VideoMME. This is likely because such tasks typically require a holistic understanding of the entire video to reach a conclusion. In these global-level

queries, our method appropriately predict relevant content spans with broader temporal coverage, often encompassing nearly the entire video. As a result, the subsequent keyframe selection reduces to uniform sampling over the whole video, yielding comparable performance. This observation highlights a specific scenario where our approach converges with the baseline.

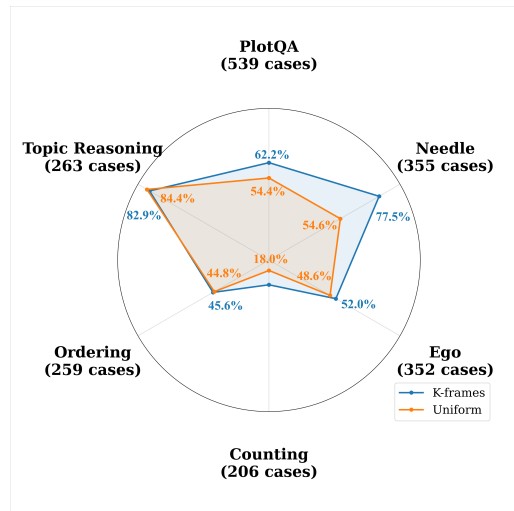

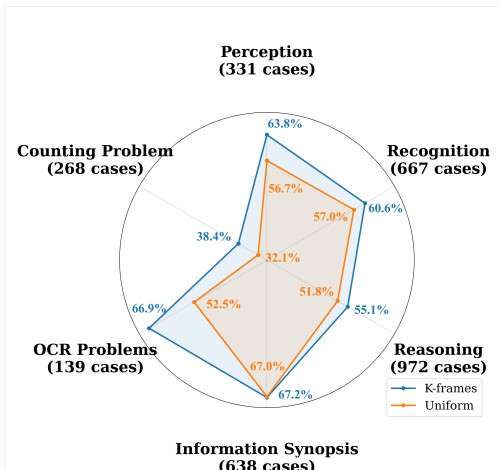

Figure 15: Performance on some MLVU subtasks. The downstream model is Qwen2.5-VL-7B with frames $k = 8$.

Figure 16: Performance on VideoMME subtasks. The downstream model is Qwen2.5-VL-7B with frames $k = 8$.

E.2 MORE ABLATION ANALYSIS

Table 10: Ablation study on the utility of including K-frames' generated reason text in the downstream model's prompt. "+ ours" refers to using our keyframe selection method. "+ ours*" indicates that in addition to using our selected frames, the textual reason for each clip's selection was also included in the prompt for the downstream model.

| Models | Size | Frames | MLVU | VideoMME | | | | LVBench |
|---|---|---|---|---|---|---|---|---|
| | | | | Short | Medium | Long | Avg | |
| InternVL3.5 | 8B | 8 | 60.5 | 68.0 | 56.7 | 49.7 | 58.1 | 57.7 |
| + ours* | 8B | 8 | 65.6 | 67.7 | 57.8 | 47.9 | 57.8 | 52.5 |
| **+ ours** | 8B | 8 | **64.4** | **71.4** | **59.0** | **50.7** | **60.4** | **60.0** |
| InternVL3.5 | 8B | 32 | 67.0 | 75.7 | 64.3 | 53.9 | 64.6 | 60.1 |
| + ours* | 8B | 32 | 65.3 | 68.0 | 58.3 | 46.9 | 57.7 | 53.8 |
| **+ ours** | 8B | 32 | **68.4** | **75.9** | **64.2** | **55.1** | **65.1** | **61.8** |

**Ablation of Including Reason Text in Downstream Prompts.** We conducted an ablation study to determine whether the textual explanations generated by K-frames for clip selection could further improve downstream task performance. To do this, we appended the reason text to the prompt given to the final downstream QA model. The results, presented in Table 10, shows: while our K-frames selection method (+ ours) significantly outperforms the baseline, including the reason text (+ ours*) degrades performance across most benchmarks.

We attribute the observed performance degradation to the design of K-frames. The selector is a lightweight MLLM whose core strength is relevance discrimination—identifying query-relevant segments—rather than accurate answer generation. Consequently, the accompanying reason text, although correctly indicating relevance, may include the selector's own preliminary reasoning or partial answers. These artifacts can introduce distracting or misleading cues that interfere with the downstream model's more sophisticated reasoning process, leading to reduced end-to-end performance.

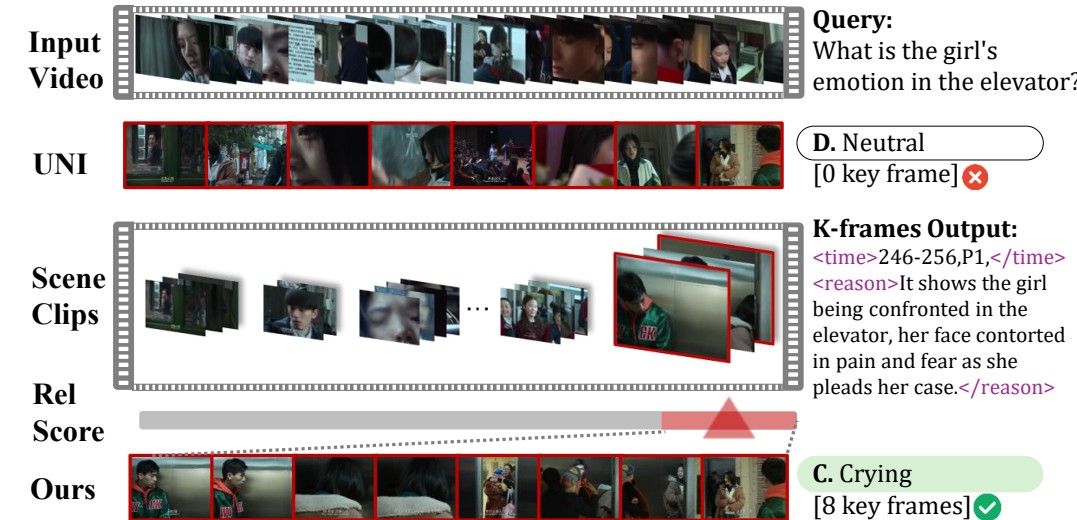

Figure 17: Qualitative comparison between uniform sampling and our K-frames method with the number of frames set to $k = 8$.

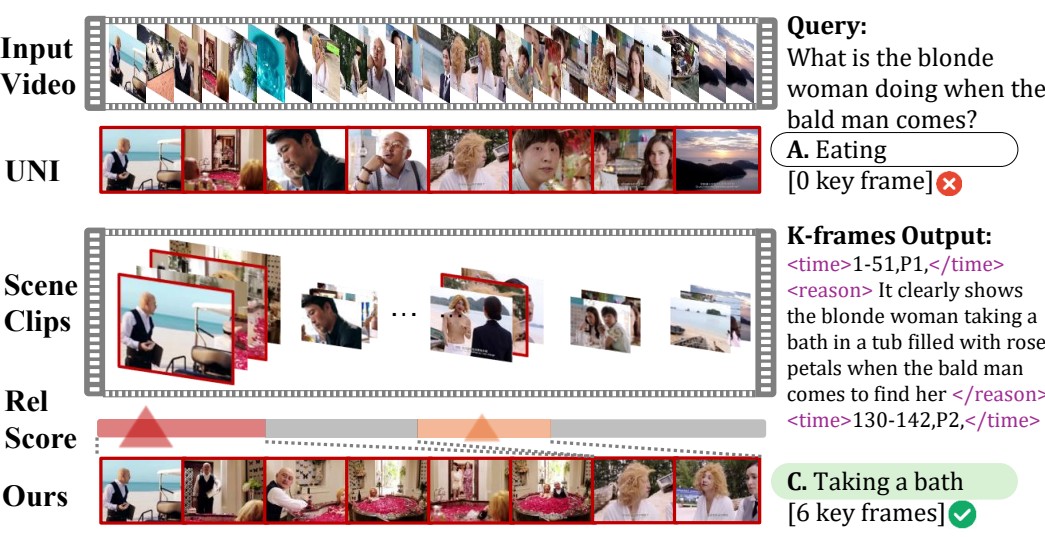

Figure 18: Qualitative comparison between uniform sampling and our K-frames method with the number of frames set to $k = 8$.

### E.3 MORE QUALITATIVE ANALYSIS

To further illustrate the robustness and interpretability of our method across different number of frame set, we provide additional qualitative comparisons in this subsection. As shown in Figure 17 and 18, when operating with the number of frame set $k = 8$, K-frames concentrates its selection entirely within the highest-scoring scene clips to capture the most critical visual evidence. When the frame set is increased to $k = 32$, K-frames showcases its flexible, multi-scale selection capability. As seen in Figure 19 and 20, our model continues to densely sample the most relevant clips, such as the man digging goods from ice or the couple surrounded by cardboard boxes. It also dynamically allocates a portion of its larger budget to select frames from other, less critical scene clips. This strategy provides a richer and more comprehensive visual context to the LLM while still prioritizing the most query-relevant information, further demonstrating the adaptability of our clip2frame paradigm.

**Query:** In the video, what did the man in black dig the goods out of ?

**Input Video:**

**Uniform Sampling:**

**Downstream Prediction:** A. Cabinet ❌ [2 key frames]

**Relevant Score**

**K-frames (Ours)**

**Downstream Prediction:** D. Ice ✅ [18 key frames]

**K-frames Output**

<time>**84-89, P2,**</time><reason>This scene provides the most critical visual evidence, showing the man in black (the thief) actively digging the goods out of a large vat filled with ice. This directly answers the question.</reason>
<time>**102-115, P1**,</time><reason>This scene shows the immediate aftermath of the digging, with the man in black struggling with the ice-covered items he just pulled out of the vat, reinforcing the answer.<reason>

Figure 19: Qualitative comparison between uniform sampling and our K-frames method with the number of frames set to $k = 32$.

**Query:** What are the couple surrounded by while sitting on the couch?

**Input Video:**

**Uniform Sampling:**

**Downstream Prediction:** A. Pillows ❌ [1 key frames]

**Relevant Score**

**K-frames (Ours)**

**Downstream Prediction:** D. Cardboard boxes ✅ [18 key frames]

**K-frames Output**
<time>84-89, P2,</time>
<reason>This scene provides a clear, direct view of the couple sitting on a couch. Around them, on the couch cushions and a pillow on the armrest, there are several cardboard boxes, which directly answers the question.</reason>

Figure 20: Qualitative comparison between uniform sampling and our K-frames method with the number of frames set to $k = 32$.

