# OpenReview forum: "K-frames: Scene-Driven Any-k Keyframe Selection for Long Video Understanding"
_ICLR.cc/2026/Conference — Submitted to ICLR 2026_

### Official Review · Reviewer_LNvJ · 2025-10-22

**Soundness:** 3
**Presentation:** 2
**Contribution:** 2
**Rating:** 4
**Confidence:** 4

**Summary:**

This paper introduces K-frames, a scene-driven frame sampling framework for long video understanding that prioritizes semantically relevant segments based on user queries. The authors construct PeakClips, a large-scale dataset with hierarchical (video/chapter/scene-level) captions and LLM-annotated query-relevance scores (P1/P2 priority). Key contributions include: (1) a three-stage pipeline—scene segmentation, hierarchical captioning, and LLM-guided relevance scoring; (2) two sampling strategies—Focused Sampling (from high-relevance clips only) and Hybrid Sampling (combining relevant clips and background); and (3) extensive experiments showing consistent gains over uniform sampling across multiple benchmarks (e.g., MLVU, VideoMME) and video-LLMs (e.g., Qwen2.5-VL, LLaVA-Video), especially under low frame budgets (e.g., +25.6% on MLVU with 8 frames). The method effectively bridges semantic scene structure with efficient visual token selection for query-conditioned video understanding.

**Strengths:**

The paper demonstrates strong originality by reframing keyframe selection as a scene-driven, query-aware process, shifting from uniform or heuristic sampling to a semantically grounded, relevance-guided paradigm. Its technical quality is high: the proposed K-frames method is well-motivated, accompanied by a carefully constructed dataset (PeakClips) with hierarchical annotations and LLM-derived relevance scores, and validated through rigorous experiments across multiple benchmarks and model scales. The clarity of presentation is excellent, with intuitive figures, clear algorithmic descriptions, and transparent reporting of design choices (e.g., P1/P2 prioritization, frame allocation). Most importantly, the work holds significant practical impact: it enables more efficient and accurate long video understanding under tight frame budgets, a common real-world constraint, while offering a model-agnostic front-end that can enhance any video-LLM without architectural changes. By aligning frame selection with semantic scene structure and query relevance, the paper advances both the methodology and infrastructure for scalable video reasoning.

**Weaknesses:**

- The article is a bit confusing in its use of symbols, font, and formulas. This is not conducive to readers' understanding of the article.

- The paper used some hyperparameters 4.3, 4.9, $w_1$, $w_2$, $\alpha$, and $\beta$ when constructing the PeakClips dataset, but lacked further explanation of the hyperparameter selection.

- Are the claims about model size in Table 1 fair? K-frames based on Qwen2.5-VL will introduce additional 3b parameters during inference and evaluation.

- The reward function design for the RL stage is not clear enough

- Lack of discussion of limitations.

- Minor Weaknesses
  - Line 97: `visual content.;` extra period
  - Line 155: $\{ clip_{i}^{v}\}=[frame\_start_{i}^{v}, frame\_end_{i}^{v}]$ can it be more simple? Moreover, the specific meaning of the superscript $i$ does not seem to be introduced.
  - Line 193: `SIGLIP similarity`, missing necessary citations
  - Line 258: `Mao et al. (2023)`, inconsistent citation format
  - Line 377: `k keyframes` -> $k$ keyframes

**Questions:**

- The proposed K-frames is coupled with MLLM (Qwen2.5-VL-7B) during the training of Stage-2(SFT) and Stage-3 (RL). Although Qwen2.5-VL-7B is frozen during training, will the performance fluctuate if other models are used?

- In Table 2, for the same 8 frames, the performance of Qwen2.5-VL-72B is lower than that of Qwen2.5-VL-7B on Needle-QA, which seems a bit counterintuitive.

- Can the authors provide a comparison of K-frames with other frame selection methods?

- The paper lacks additional computational overhead and latency associated with K-frames.

I look forward to discussions with the author during the rebuttal phase and would be happy to improve my score.

---

> ### Author Response · Authors · 2025-11-25
>
> We thank the reviewer for the detailed and careful feedback, for pointing out typos / notation issues, and for the positive assessment of the practical impact of our approach.
>
> ### **Q1. Coupling with Qwen2.5-VL-7B in RL and behavior with other downstream models**
>
> We first clarify the training pipeline:
>
> - **Stage-1 & Stage-2 (SFT):** only the 3B selector (Qwen2.5-VL-3B) is trained, using PeakClips annotations. The downstream MLLMs are not involved.
> - **Stage-3 (RL):** we freeze a downstream QA model (Qwen2.5-VL-7B in the main paper) and use it only to compute rewards; the 7B model itself is never updated. The selector remains 3B.
>
> To address the concern about coupling to a specific downstream model, we ablate the choice of reward backbone in the RL stage.
>
> **Table R5. Ablation on the reward backbone used in RL**
>
> | Reward backbone for RL | Needle-QA | MLVU M-Avg |
> |------------------------|-----------|------------|
> | Qwen2.5-VL-7B          | 71.6      | 56.6       |
> | Intern3.5-VL-8B         | 71.8      | 56.7       |
>
> We observe that using different reward backbones leads to similar improvements over uniform sampling, and stronger reward backbone may improve the performance further. Qwen2.5-VL-7B provides a good trade-off between reward quality and RL cost. Importantly, once trained, the deployed selector is always the same 3B model, and can be plugged into other downstream MLLMs without retraining.
>
>
> ### **Q2. Why does Qwen2.5-VL-7B perform comparably to 72B on Needle-QA?**
>
> We agree that this looks counter-intuitive. Our understanding is:
>
> Needle-QA constructs each example by randomly inserting a short “needle” segment containing evidence into a longer background video and annotating a corresponding pair of questions and answers. It becomes easy once the correct clip is localized. Questions are typically simple, so a 7B model already answers most examples correctly once provided with the right frames.
>
> We emphasize that **K-frames benefits both 7B and 72B models** for long-video understanding.
>
>
> ### **Q3. Comparison with other frame selection methods**
>
> The new comparison study (Table R1, R3) directly addresses this concern.
>
> ### **Q4. Computational overhead and latency of K-frames**
>
> To directly address this question, we include an inference-time study (Table R4 in the rebuttal).
>
>
> ### **W1. Clarity of symbols, formulas, and notation**
>
> We appreciate the reviewer’s comments on clarity. We carefully proof-read the entire manuscript and fixed all reported typos.
>
>
> ### **W2. Model size & “Size” column in Table 1**
>
> In all main tables (including Table R1), the “Size” column refers to the number of parameters of the **downstream QA model** that produces the final answers, following the convention in ViaRL. For training-free selectors such as VideoTree, the selection stage may rely on ChatGPT-style models to select keyframes, but the evaluation model can be either a small open-source MLLM or ChatGPT itself.
>
> In Table R1 we explicitly report the size / type of the model that actually **answers the benchmark questions**.
>
>
> ### **W3. Reward design and explanation**
>
> We adopt the **GRPO** algorithm for RL and only trained with multi-choice question. For each prompt, we roll out the policy multiple times (N = 4) to obtain several candidate K-frames predictions.
>
> For each candidate:
>
> 1. We uniformly sample 16 keyframes only from the predicted relevant clips.
> 2. We feed these frames with the question into the frozen downstream Qwen2.5-VL-7B model to obtain an answer.
> 3. We then compute the group‑wise advantage by comparing the output log‑probability of the ground‑truth letter with the average log‑probability of the other options, i.e.
>
> $$r = \log(p_{\text{ans}=y}) - \log\left(\frac{1}{N}\sum p_{\text{ans}\neq \hat y}\right)$$
>
> 4. We apply a temperature \(\tau\) and a squashing function:
>
> $$\text{reward} = \tanh\left(\frac{1}{\tau} r\right)$$
>
> - The temperature adjusts sensitivity to log-ratio changes.
> - The $\(\tanh(\cdot)\)$ bounds the reward, keeping magnitude and variance well-controlled for stable GRPO training.
>
> ### **W4. Limitations (now explicitly discussed)**
>
> We acknowledge the original draft did not clearly discuss limitations. We will add the following paragraph:
>
> - Because the selector is based on Qwen2.5-VL-3B with a fixed input of 256 frames, our deployment may still be sparse for extremely long videos (e.g., > 2 hours). Handling ultra-long videos may require hierarchical or streaming variants.
> - Our method is scene-driven: it works better for long videos with multiple distinct scenes and extended temporal structure. For  short videos with little scene change, dense retrieval or frame-combination methods may be more effective. We will mention this as a scenario where K-frames is less advantageous.

---

> ### Comment · Reviewer_LNvJ · 2025-11-26
>
> Thanks for the author's detailed response. The authors have responded to most of my questions, but I still have some concerns.
>
> 1. As I mentioned in Weekness, the paper used some hyperparameters, including some magic numbers (4.3, 4.9) when constructing the PeakClips dataset, but lacked further explanation and illustration of the hyperparameter selection.
>
> 2. Regarding the declaration of model size in Table 1, the authors claim to have followed the convention in ViaRL[1]. I only found a preprint version of this paper, and it seems that ViaRL has not yet been reviewed and accepted. I still maintain that the authors should **add the number of parameters introduced by the K-frame itself to the open-source MLLM** when making comparisons, so readers can better understand the K-frame's contribution.
>
> 3. The current PDF document's description of the reward is **still** not detailed enough (It's unclear whether the author has updated the PDF.). Furthermore, the authors provide detailed training objectives for the SFT stage in Equation 1; providing similar objectives for the GRPO stage can improve the reading experience and consistency in article representation.
>
> 4. The Limitations section and Table R4 should be added to the manuscript.
>
> [1] Xu Z, Dai Q, Xie T, et al. ViaRL: Adaptive Temporal Grounding via Visual Iterated Amplification Reinforcement Learning[ J]. arXiv preprint arXiv:2505.15447, 2025.
>
> [2] Wang Z, Yu S, Stengel-Eskin E, et al. Videotree: Adaptive tree-based video representation for llm reasoning on long videos[C]//Proceedings of the Computer Vision and Pattern Recognition Conference. 2025: 3272-3283.
> ***
> Some friendly advise:
> - The author cited several articles[1,2] in their response; providing necessary citations helps reviewers quickly find the relevant content.

---

> > ### Author Response · Authors · 2025-11-28
> >
> > We thank the reviewer again for the follow‑up comments and the very concrete suggestions.
> >
> > ### **(1) Hyperparameters for PeakClips (4.3 and 4.9)**
> >
> > As described in Sec. 3.1 and App. C.1, PeakClips is built in two steps:
> >
> > 1.  LLM score (1–5):
> >     Gemini 2.5 Pro rates each scene with an ordinal score and a rationale, where
> >     *   5 = directly relevant / contains decisive evidence
> >     *   4 = highly relevant / strong contextual support
> >     *   3/2/1 = progressively weaker or no relevance.
> > 2.  Similarity refinement:
> >     For each scene, we compute the average SIGLIP similarity between the query (question + answer) and all frames in that scene. The final clip score is a weighted combination of LLM score and SIGLIP similarity. This keeps the semantic judgment of the LLM and refines cases using visual similarity: scenes with very high similarity are pushed up, those with low similarity are pushed down.
> >
> > **Why 4.9 and 4.3?**
> >
> > Our goal is to distinguish:
> > *   P1 (top‑priority): scenes that almost certainly contain the decisive evidence for answering the query. final score ≥ 4.9
> > *   P2 (secondary‑priority): scenes that strongly support the answer (setup, partial evidence, complementary views), but are slightly less critical. 4.3 ≤ final score < 4.9
> >
> > The intuition is:
> > *   `4.9` singles out scenes that have an LLM score of 5, or have an LLM score of 4 but exhibit very high frame–query similarity.
> >     *   A clip originally scored 4 but with very high similarity can be promoted to P1.
> > *   `4.3` is chosen so that most clearly supportive 4‑level scenes are kept as P2.
> >     *   A clip scored 4 with low similarity will fall below 4.3 and not even be P2.
> >
> > The two numbers were selected manually. We further show the score distribution in Figure. 4(c).
> >
> > **How much do P1/P2 matter in practice?**
> >
> > As described in Appendix D.2, in the main experiments Table 1 and Table R1 with 8 frames we use Focused Sampling: all frames are drawn uniformly from predicted P1/P2 clips. For 32/64 frames, we use Hybrid Sampling, dynamically mixing frames from predicted clips and the background with a default weight ratio of 4:1 (predicted : non‑predicted). In the main table, P1 and P2 share the same weight, so P1/P2 tagging is not yet fully exploited.
> >
> > In the revision, we add an ablation where we set P1:P2:background = 4:2:1, which improves performance compared to the uniform treatment of P1/P2:
> >
> > | Model                     | Frames | Priority tag | Needle‑QA | MLVU M‑Avg |
> > |---------------------------|:------:|:-----------:|----------:|-----------:|
> > | Qwen2.5‑VL 7B + K‑frames |   32   |      ❌      |    79.4   |    65.9    |
> > | Qwen2.5‑VL 7B + K‑frames |   32   |      ✅      |    79.5   |    66.1    |
> >
> > This shows that the P1/P2 tags capture useful fine‑grained relevance information and enable users to customize their own any‑k sampling strategies for different tasks.
> >
> > ### **(2) Model size and the parameters introduced by K-frames**
> >
> > Our intention in the “Size” column of Table 1 and Table R1 is to show the variant capability of the downstream QA model. We agree this can be ambiguous; in the revision we now explicitly show that K-frames introduces an additional 3B selector on top of the open-source downstream MLLM so that readers can see both the QA model size and the extra parameters brought by our method.
> >
> > ### **(3) Updated paper**
> >
> > We apologize that the PDF was not updated in time during the rebuttal. We now update the PDF accordingly (new changes are highlighted in red in the revised manuscript).
> >
> > ### **(4) Citations in responses**
> >
> > We appreciate the reviewer’s friendly advice on providing full citations for referenced methods. In the revised paper and author response, we have also included the corresponding references so that reviewers can quickly locate the related content.

---

### Official Review · Reviewer_RZLy · 2025-10-25

**Soundness:** 3
**Presentation:** 3
**Contribution:** 2
**Rating:** 4
**Confidence:** 4

**Summary:**

This paper proposes K-Frames, a _scene-driven_ and _query-aware_ framework for keyframe selection in long-video understanding.
The method first constructs a _scene similarity graph_ from frame-level embeddings, connecting visually and temporally adjacent nodes. Node salience is then scored using a combination of _query relevance_ and _diversity penalties_. Top-K keyframes are sampled from the most salient scene nodes. Additionally, the authors introduce a new dataset, **PeakClips** (200K video-query pairs), annotated with scene boundaries and hierarchical captions, and train their model through a three-stage curriculum: two supervised fine-tuning stages followed by a reinforcement-learning (RL) stage that aligns clip selection with downstream QA accuracy. Experiments on VideoMME, MLVU, and LVBench demonstrate consistent improvements over uniform sampling and recent baselines (e.g., ViaRL).

**Strengths:**

- **Graph-based formulation.** The idea of modeling temporal relations among scenes as a graph is conceptually appealing and promotes global consistency in keyframe selection.

- **Balanced relevance–diversity scoring.** The integration of query relevance with diversity control is reasonable and supported by formal design.

- **Extensive experiments.** Evaluation covers multiple benchmarks (MLVU, VideoMME, LVBench) and both open- and closed-source MLLMs (QwenVL, Gemini, GPT-4o), demonstrating cross-model generality.

- **Dataset contribution.** The PeakClips dataset provides a large-scale resource for scene-level grounding and may be useful for future works on temporal reasoning.

**Weaknesses:**

- **Limited conceptual novelty.** The combination of scene segmentation, query-aware relevance scoring, and diversity optimization has been widely explored in recent works. Beyond earlier methods such as Frame-Voyager (Yu et al., 2024), AKS (Tang et al., 2025), and EFS (Sun et al., 2025), several newer approaches—T* (Ye et al., 2025), Logic-in-Frames (Guo et al., 2025), mDP3 (Sun et al., 2025), VSI (He et al., 2025), and Nar-KFC (Fang et al., 2025)—already integrate scene- or event-level temporal structuring with query-conditioned frame retrieval and diversity-aware selection. Compared to these, K-Frames largely reformulates the same paradigm through a graph abstraction and an added SFT→RL curriculum, without introducing fundamentally new principles of temporal reasoning or representation learning.
Overall, the contribution appears incremental rather than conceptual, as many of the above methods already address temporal coherence, multi-scale selection, and reasoning alignment more systematically.

- **Questionable contribution of the graph structure.** The paper does not convincingly show why graph-based modeling is superior to simpler alternatives such as clustering-based grouping or temporal-adjacency filtering. There is no ablation isolating the graph connectivity or node-edge design—so it remains unclear whether the graph actually improves selection quality beyond standard grouping.

- **Heuristic-heavy and weakly learned pipeline.** Despite being presented as a learning framework, most components rely on fixed heuristics:
Scene boundaries from histogram differences, Relevance computed via frozen Gemini 2.5 Pro + SIGLIP, Hand-tuned thresholds for P1/P2 classification. The RL stage fine-tunes only a small policy head and uses rule-based rewards. Consequently, the system remains largely training-free and non-adaptive to new domains or tasks.

- **Dataset quality and scalability concerns.** The PeakClips dataset is automatically labeled via LLM prompts, which raises reproducibility and reliability concerns. No human verification or inter-annotator analysis is reported. Moreover, since PeakClips depends heavily on proprietary Gemini outputs, its release or reproducibility for academic use is questionable.

- **Marginal improvement over strong baselines.** Although Table 1 shows large gains over uniform sampling, this baseline is weak.
When compared to recent training-based methods (e.g., ViaRL, EFS, BOLT), the gains are modest (≈ 2–4 points). It is unclear whether the added complexity of dataset construction and RL fine-tuning is justified by these limited improvements.

- **Terminology mismatch.** The term “Scene Graph” is misused: the proposed structure lacks entities, relations, or semantic predicates typically associated with scene graphs in computer vision. A more accurate term would be “temporal-visual similarity graph.”

- **Evaluation fairness and transparency.** The reported results use proprietary models (Gemini 2.5 Pro, GPT-4o) for scoring and evaluation.
It is unclear how the model’s outputs were filtered or whether any cherry-picking occurred. Furthermore, no statistical significance or variance across runs is provided.

- **Computational efficiency not demonstrated.** The paper claims “any-K sampling and improved efficiency,” but no profiling or runtime analysis is given. In practice, the three-stage pipeline (scene segmentation → SFT → RL) may be substantially more expensive than simple similarity-based retrieval.

**Questions:**

1. **Role and stability of RL fine-tuning.** How exactly does the RL stage affect selection quality compared with SFT-only training (Table 3)? Are the observed improvements statistically significant and consistent across random seeds?
2. **Graph formulation justification.** Could the authors demonstrate that the proposed graph structure yields measurable improvement over simpler temporal clustering or adjacency-based grouping? What concrete advantage does the “scene graph” abstraction provide beyond naming?
3.  **Generalization and compositional reasoning.** How robust is the system when handling compositional or cross-scene queries beyond single-event localization, and does the model generalize to non-QA tasks such as captioning or temporal reasoning?
4. **Dataset reliability and openness.** Since PeakClips is auto-labeled using Gemini 2.5 Pro, can the dataset be publicly released under license, and how was its annotation quality or bias verified?
5. **Interpretability and rationale faithfulness.** The paper highlights interpretable rationale outputs—can the authors quantify or evaluate their faithfulness, e.g., via human evaluation or alignment with visual evidence?
6. **Efficiency and computational overhead.** Please report the end-to-end runtime and computational cost (scene segmentation + SFT + RL) relative to uniform or retrieval-based baselines, especially when scaling to long videos.

---

> ### Author Response · Authors · 2025-11-25
>
> We appreciate the time taken to review our work. However, we must explicitly address a fundamental factual error in the review regarding the core methodology of our paper. The review repeatedly describes our method as "graph-based" referencing specific components ("scene similarity graph", "node salience", "diversity penalties") that **DO NOT EXIST** in our manuscript. Our method **K-frames is a scene-driven keyframe selector, not a graph-based retrieval algorithm**.
>
>
> ### **(1) FUNDAMENTAL MISUNDERSTANDING: K-Frames is NOT a "Graph-Based" Method**
>
> - **K‑frames is a scene‑driven keyframe selector** built on Qwen2.5‑VL‑3B, which segments videos into different scenes by directly predicting start/end timestamps and relevance score (clip2frame prediction) based on the video and query (Section 3, Figure 1, Figure 3).
> - We **do not construct a graph** over scenes, nor introduce edges or graph connectivity losses. We do **not** employ "diversity penalties" or "adjacency-based filtering". Instead, we use a learned policy to predict informative clips.
>
> ### **(2) Conceptual novelty and relation to prior work**
>
> The review asserts that our work is a "reformulation of the same paradigm through a graph abstraction". Since we do **not use the "graph abstraction"** which is claimed we do, the argument that we are merely adding a graph to existing methods is unfounded. Below is a clear clarification of novelty to prior keyframe selection work.
>
> We first introduce the two major paradigms of existing keyframe selection methods:
>
> 1. **Text–frame retrieval**: rank frames independently by similarity to the query, ignoring temporal continuity.
> 2. **RL‑based frame subset optimization**: directly optimize which frames to keep; selections are typically sparse and temporally disjoint.
>
> **K‑frames introduces a different perspective**:
> - K-frames is reframing keyframe selection as a **clip2frame prediction**, where the model first predicts query‑conditioned temporally contiguous clips with rationales and priority tags, and only then samples any‑k keyframes from these clips, rather than the traditional retrieval (ranking via similarity) or combinatorial optimization (RL search) paradigms.
> - Unlike retrieval methods that treat frames as independent units, this clip-first design naturally preserves scene continuity, supports any‑k budgets, and gives interpretable outputs (segments + reasons).
>
> The new comparisons in Table R1 show that this paradigm allows a single 3B selector to substantially boost a wide range of MLLMs (Qwen, InternVL, GPT‑4o, Gemini) on long‑video benchmarks, which we believe is a meaningful conceptual and practical contribution beyond simply adding more heuristics.
>
> ### **(3) Contribution and cost of RL stage**
>
> We emphasize that the RL stage is **not** an auxiliary or cosmetic addition, but a principled alignment step that directly optimizes clip selection for downstream reasoning quality.
>
> - Table 3 in the paper already shows that RL yields a clear performance improvement of **1.4% (MLVU M‑Avg)** and **3.1% (Needle‑QA)** over the SFT‑only model.
> - A more explicit discussion of **Cost vs. Benefit**: the RL stage takes ~40 hours on our hardware (with the selector only), but once trained, the selector is reused across all downstream models without additional training.
>
> We also emphasize that unlike ViaRL, which requires an iterative, joint training of the 7B downstream model (expensive and intrusive), we do not update the downstream 7B/72B model during RL; it is only used to compute rule‑based rewards, so the cost stays bounded.

---

> > ### Author Response · Authors · 2025-11-25
> >
> > ### **(4) Heuristics for segmentation and relevance scoring**
> >
> > There appears to be a confusion between the dataset construction pipeline and the training pipeline. We would like to clarify that K‑frames is trained end-to-end (fully learned) through a three-stage curriculum. Any heuristic methods are applied **only during offline dataset construction** and are not part of the model's runtime inference logic.
> >
> > Specifically, we employ a three-stage pipeline (illustrated in Figure 2) to annotate the PeakClips dataset with relevance scores:
> >
> > - **Scene Segmentation**: We first detect scene boundaries using a standard histogram‑difference boundary detector [1,2], which is subsequently refined and optimized using Gemini 2.5 Pro.
> > - **Relevance Scoring**: Relevance scores are derived by combining structured judgments from Gemini 2.5 Pro (via carefully designed prompts) with visual similarity computed by SIGLIP. The final score is a weighted fusion of the LLM-based assessment and the visual similarity measure.
> >
> > To enhance transparency and reproducibility:
> >
> > - We provide the full set of prompts in Appendices C and D, along with example annotated scenes in Figures 9–10.
> > - We include detailed statistics on score distributions across sources (Table 6, Figure 4), capturing inter-dataset variations and robustness.
> > - An ablation study demonstrates that training K‑frames without SIGLIP refinement leads to decreased performance, confirming the empirical value of the combined heuristic.
> >
> > [1] Gushchin, A., Antsiferova, A., & Vatolin, D. (2021). *Shot Boundary Detection Method Based on a New Extensive Dataset and Mixed Features*. arXiv preprint arXiv:2109.01057.
> > [2] Kar, T., et al. (2024). *Video Shot-Boundary Detection: Issues, Challenges and Solutions*. Artificial Intelligence Review, 57(4), 104.
> >
> > ### **(5) Dataset quality, openness, and potential bias**
> >
> > We disagree with implication that our PeakClips dataset is of low quality or not reproducible. We take dataset quality seriously:
> >
> > - We use state‑of‑the‑art Gemini 2.5 Pro for structured captioning and relevance scoring with explicit reasons, which facilitates automatic sanity checks.
> > - We report comprehensive statistics (video duration, scenes per video, score distribution) and will add manual spot‑check summaries (agreement with human annotators) to strengthen the quality claim.
> > - We will release **PeakClips under an open license**, including prompts and scripts, so others can inspect, critique, or extend it.
> >
> > PeakClips is a well-justified, reconstructable resource with **dense annotations** (event timestamps, relevance scores, and textual rationales), making it a versatile resource for broader tasks such as temporal localization, video summarization, and fine-grained video understanding.
> >
> > ### **(6) Evaluation fairness**
> >
> > We assure that all comparisons in our paper were conducted with strict fairness:
> >
> > - We used a standardized evaluation framework (**VLMEvalKit**) to test our method under identical conditions.
> > - Evaluation was done across three widely-used **public long-video benchmarks**: VideoMME, MLVU, and LongVideoBench (LVBench).
> >
> > Scores reported (Table 1, 2, 3, 4) come from the **unified evaluation process**. The resulting performance gains of K-frames are consistently measured against the exact same protocol, same test queries and metrics as the baselines.
> >
> > ### **(7) Computational efficiency and comparison to simple baselines**
> >
> > The concern about computational efficiency is well-taken. Crucially, we need to first highlight **the difference between Training Time and Inference Time: SFT and RL stages occur only once during training**. At inference, the selector is a compact 3B model that performs fast clip prediction without any RL rollouts or re-evaluation loops.
> > The new computational results (Table R4) demonstrate this advantage:
> > K-frames is competitive or faster than LanguageBind while providing higher accuracy.
> > It is significantly more efficient than VideoTree, whose pipeline spends excessive time on caption generation and repeated GPT-reasoning passes over the video.
> > Our method offers a superior trade-off between performance and runtime cost.

---

### Official Review · Reviewer_kpwF · 2025-11-01

**Soundness:** 2
**Presentation:** 3
**Contribution:** 3
**Rating:** 6
**Confidence:** 4

**Summary:**

This paper proposes K-frames, a scene-driven approach for keyframe selection in long video understanding. Instead of selecting individual frames, it reframes the task as "clip-to-frame" prediction: first identifying query-relevant video segments, then sampling keyframes from these coherent clips. This addresses issues like temporal discontinuity and inflexibility in prior methods.

To support this, the authors construct PeakClips, a large-scale dataset with 200K query-conditioned annotations, and train the model using a three-stage curriculum (two SFT stages + RL). Experiments show K-frames works as a plug-and-play front-end that boosts various MLLMs' performance on long-video benchmarks.

**Strengths:**

Clip-level selection: Choosing coherent segments first ensures temporal continuity.

Valuable dataset: PeakClips provides rich annotations useful for broader research.

Model-agnostic: Consistently improves various models (Qwen, GPT-4o, Gemini) without modification.

**Weaknesses:**

Missing key comparisons: Lacks direct comparison with advanced temporal search methods (e.g., VideoAgent, T*, VideoTree).

Indirect validation: Relies solely on downstream QA accuracy for evaluation, without directly measuring clip selection quality (e.g., overlap with ground-truth segments).

Unclear overhead: Fails to report the additional time cost of the "clip-finding" step, leaving practical efficiency in question.

**Questions:**

See Weaknesses

---

> ### Author Response · Authors · 2025-11-25
>
> We appreciate the reviewer's clear summary and for highlighting our strengths (clip‑level selection, dataset value, and model‑agnostic improvements). We respond to the weaknesses below.
>
> ### **(1) Comparisons to advanced temporal search methods**
>
> The new **comparison study (Table R1, R3)** directly addresses this concern.
>
> ### **(2) Direct quality evaluation of clip selection (not only QA accuracy)**
>
> We agree that evaluating only downstream QA accuracy makes it harder to disentangle improvements due to better selection vs. downstream reasoning. To address this, we compute overlap between predicted P1/P2 clips and top‑relevance scenes (IoU / precision /recall) on a held‑out subset.
>
> | Method    | IoU   | Precision | Recall |
> |-----------|-------|-----------|--------|
> | K‑frames  | 44.8  | 60.3      | 64.3   |
>
> ### **(3) Overhead and practical efficiency**
>
> The new **inference‑time study (Table R4)** directly addresses this concern:
>
> - On MLVU (930 s average length), K‑frames improves accuracy from 52.3→60.4 over LanguageBind while having slightly improved latency (10.6 vs 11.2 seconds).
> - Compared with VideoTree, the overhead of caption generation and repeated LLM queries dominates: VideoTree needs >24.3s per long video, while K‑frames finishes in ~10–13s with comparable accuracy.

---

### Official Review · Reviewer_B5co · 2025-11-01

**Soundness:** 3
**Presentation:** 3
**Contribution:** 2
**Rating:** 4
**Confidence:** 4

**Summary:**

This paper proposes K-frames, a scene-driven keyframe selection paradigm that selects coherent query-relevant clip rather than isolated frames, enabling flexible “any-k” frame budgets. The method is trained with a progressive 3-stage curriculum (two SFT stages + RL) using PeakClips, a newly constructed 200K query-conditioned highlight dataset. Experiments on long-video benchmarks show that K-frames yields more effective, interpretable, and plug-and-play keyframe selection across scales.

**Strengths:**

This paper proposes a novel key-frame sampling method. It explicitly considers scene structure in videos rather than treating them as simple independent frames — this is a very good starting motivation. In addition, the dataset construction is comprehensive and well designed.

**Weaknesses:**

1.What is the advantage of scene-based splitting compared to directly segmenting the video into equal-length intervals?

2.Unlike traditional frame-level semantics, “K-FRAMES” selects key frames based on complete temporal structure. This advantage should manifest in tasks such as action recognition and temporal reasoning tasks that depend on continuous dynamics rather than static appearance. Can the authors provide results on such tasks?

3.The authors train an additional module (Qwen-2.5-VL 3B) for key-frame sampling. I am curious how its performance compares to off-the-shelf multimodal retrievers such as LanguageBind.

4.In Section 4.3, the temporal localization experiments are not very convincing because 'Needle QA' is too easy. Consider adding stronger evidence (more challenging temporal localization proof).

5.The “Priority Tag” idea is interesting, but the authors do not discuss it in depth. Could the authors demonstrate whether this design is necessary with ablation experiments?

**Questions:**

If the authors can answer the questions raised in the Weaknesses section, I will consider raising my score.

---

> ### Author Response · Authors · 2025-11-25
>
> We thank the reviewer for the positive assessment of our keyframe sampling paradigm and PeakClips dataset.
>
>
> ### **(1) Advantage of scene‑based splitting vs equal‑length segmentation**
>
> Long videos naturally consist of semantically coherent scenes with varying lengths. In practice, K‑frames can adaptively perceive scene boundaries and group frames with consistent content semantics into a single scene. By contrast, the window length of equal-length windows is a hyperparameter. Using the same window length for videos with different content and varying lengths is inappropriate.
>
> In essence, **K‑frames adaptively partitions windows based on the semantic content of the video and its relevance to the query**, instead of relying on a manually chosen, fixed temporal stride.
>
> ### **(2) Effect on other tasks**
>
> MLVU, VideoMME, and LVBench are **multi‑task long‑video benchmarks** that include temporal localization, temporal reasoning, causal reasoning, anomaly detection, action recognition, etc. As shown in **Figure. 15 & 16**, K‑frames improves not only Needle‑QA but most subtasks, especially those that require localizing fine‑grained evidence in long contexts.
>
> To further address this point:
>
> - We add experiments on **EgoSchema** and **NextQA** (Table R3), which focus on temporal reasoning in egocentric or multi‑step scenarios.
> - Qualitative visualizations (Fig. 1, 17–20) show that our predicted clips consistently align with complex queries (e.g., event ordering, object interactions), not just simple localization.
>
>
> ### **(3) Comparison to LanguageBind and other methods**
>
> The new comparison study (Table R1, R3) directly addresses this concern.
>
>
> ### **(4) Stronger temporal localization evidence beyond Needle‑QA**
>
> We agree that Needle‑QA becomes relatively straightforward once the relevant moment is localized. Figure.1 presents an example of multi‑event sequencing. As can be observed, our model accurately identifies the scenes corresponding to the four events and extracts the relevant keyframes.
>
> Our method is not specifically designed for temporal localization; instead, it predicts scenario segments relevant to the question and then selects keyframes. For a single query, multiple segments may be predicted, which cannot be directly applied to a standard localization task. Further fine‑tuning would be required.
>
> We fully endorse your idea—**extending our method to a dedicated temporal localization task will be a promising future direction**. To strengthen the current submission, we also include a direct evaluation of clip overlap with our PeakClips relevance annotations (IoU / precision / recall) to show that predicted P1/P2 clips align well with high‑relevance scenes:
>
> | Method   | IoU | Precision | Recall |
> |----------|----:|----------:|------:|
> | K‑frames | 44.8 | 60.3 | 64.3 |
>
> ---
>
> ### **(5) Priority tags and ablation on P1/P2**
>
> Thank you for pointing out the importance of the priority design. As described in Appendix D.2, in the main experiments with 8 frames we use **Focused Sampling**: all frames are drawn uniformly from predicted P1/P2 clips. For 32/64 frames, we use **Hybrid Sampling**, dynamically mixing frames from predicted clips and the background with a default weight ratio of 4:1 (predicted : non‑predicted). In the main table, P1 and P2 share the same weight, so P1/P2 tagging is not yet fully exploited.
>
> In the revision, we add an ablation where we set **P1:P2:background = 4:2:1**, which improves performance compared to the uniform treatment of P1/P2:
>
> | Model                     | Frames | Priority tag | Needle‑QA | MLVU M‑Avg |
> |---------------------------|:------:|:-----------:|----------:|-----------:|
> | Qwen2.5‑VL 7B + K‑frames |   32   |      ❌      |    79.4   |    65.9    |
> | Qwen2.5‑VL 7B + K‑frames |   32   |      ✅      |    79.5   |    66.1    |
>
> This shows that the P1/P2 tags capture useful fine‑grained relevance information and enable users to **customize their own **any‑k** sampling strategies for different tasks**.

---

### Author Response · Authors · 2025-11-25
**Reply to all Reviewers**

We thank all reviewers for their careful reading and constructive feedback.

### **Summary of New Experiments**

- **More comparisons and benchmarks.** We present comprehensive comparisons against LanguageBind (a retrieval-based method), ViaRL, VideoTree, Video-XL, and Kangaroo across long-video benchmarks. Besides MLVU, VideoMME, and LVBench, we add experiments on EgoSchema and NextQA, which focus more on temporal reasoning and egocentric understanding.
- **Efficiency overhead.** We add inference-time analysis comparing the K-frames with LanguageBind, and VideoTree.

---

> ### Author Response · Authors · 2025-11-25
> **Reply to all Reviewers**
>
> ### **New Comparison on Long-Video Benchmarks**
> **Table R1. Long-video understanding: more comparisons**
>
> | Model                 | Size | Frames | MLVU Needle | MLVU M-Avg | VideoMME Short | VideoMME Med. | VideoMME Long | VideoMME Avg | LVBench |
> |----------------------|:----:|:------:|:-----------:|:----------:|:--------------:|:-------------:|:-------------:|:------------:|:-------:|
> | **Video MLLMs**      |      |        |             |            |                |               |               |              |         |
> | VideoChat2           | 7B   | 16     | –           | 44.5       | 48.3           | 37.0          | 33.2          | 39.5         | –       |
> | VideoLLaVA           | 7B   | 8      | –           | 47.3       | 45.3           | 38.0          | 36.2          | 39.9         | –       |
> | Video-CCAM           | 14B  | 96     | 73.2        | 63.1       | 62.2           | 50.6          | 46.7          | 53.2         | –       |
> | Video-XL             | 7B   | 128    | 73.8        | 64.9       | 64.0           | 53.2          | 49.2          | 55.5         | –       |
> | Kangaroo             | 7B   | 64     | –           | –          | 66.1           | 55.3          | 46.7          | 56.0         | –       |
> | VideoTree            | –    | –      | –           | 60.4          | 67.8              | 59.9             | 54.2          | 60.6            | –       |
> | ViaRL                | 3+7B   | 8      | 73.5        | 58.2       | 65.1           | 56.1          | 50.8          | 57.3         | –       |
> | Qwen2.5-VL           | 7B   | 8      | 58.6        | 53.9       | 61.7           | 50.6          | 46.9          | 53.0         | 52.8    |
> | + LanguageBind       | 7B   | 8      | 51.6        | 52.3       | 54.3           | 49.2          | 45.9          | 49.8         | 52.2    |
> | **+ ours**| 3+7B   | 8      | **77.5 (+18.9)** | **60.4 (+6.5)** | 68.9 | 55.3 | 47.9 | **57.4 (+4.4)** | **57.7 (+4.9)** |
> | Qwen2.5-VL           | 7B   | 32     | 63.4        | 61.7       | 71.8           | 60.8          | 50.1          | 60.2         | 59.3    |
> | + LanguageBind       | 7B   | 32     | 79.0        | 64.0       | 61.7           | 55.2          | 49.0          | 55.3         | 55.2    |
> | **+ ours**| 3+7B   | 32     | **79.4 (+16.0)** | **65.9 (+4.2)** | 74.1 | 61.4 | 51.7 | **62.1 (+1.9)** | **60.5 (+1.2)** |
> | Qwen2.5-VL           | 7B   | 64     | 67.7        | 65.6       | 73.9           | 62.3          | 52.2          | 62.8         | 59.9    |
> | **+ ours**| 3+7B   | 64     | **78.9 (+11.2)** | **67.8 (+2.2)** | 75.9 | 63.7 | 53.9 | **64.5 (+1.7)** | **61.6 (+1.7)** |
> | Qwen2.5-VL           | 72B  | 8      | 51.6        | 56.3       | 65.6           | 56.6          | 51.1          | 57.7         | 55.6    |
> | **+ ours**| 3+72B  | 8      | **77.2 (+25.6)** | **63.3 (+7.0)** | 70.2 | 58.9 | 52.8 | **60.6 (+2.9)** | **59.3 (+3.7)** |
> | Qwen2.5-VL           | 72B  | 32     | 67.3        | 64.0       | 74.3           | 63.4          | 58.1          | 65.3         | 60.8    |
> | **+ ours**| 3+72B  | 32     | **78.3 (+11.0)** | **67.6 (+3.6)** | 75.2 | 66.0 | 57.8 | **66.3 (+1.0)** | **63.2 (+2.4)** |
> | Gemini 2.5 Pro       | –    | 8      | 43.4        | 54.2       | 77.7           | 67.4          | 62.1          | 69.1         | 57.8    |
> | **+ ours**       | –    | 8      | **71.6 (+28.2)** | **56.6 (+2.4)** | 79.7 | 67.2 | 62.8 | **70.0 (+0.9)** | **62.2 (+4.4)** |
> | Gemini 2.5 Pro       | –    | 32     | 74.6        | 66.0       | 87.1           | 74.9          | 69.6          | 77.2         | 64.2    |
> | **+ ours**       | –    | 32     | **80.9 (+6.3)** | **69.0 (+3.0)** | 87.1 | 76.1 | 70.9 | **78.0 (+0.8)** | **67.0 (+2.8)** |
> | GPT-4o               | –    | 8      | 58.3        | 55.4       | 67.2           | 58.6          | 53.5          | 59.7         | 49.4    |
> | **+ ours**       | –    | 8      | **75.2 (+16.9)** | **60.5 (+5.1)** | 72.4 | 60.8 | 54.6 | **62.6 (+2.9)** | **54.5 (+5.1)** |
> | GPT-4o               | –    | 32     | 71.3        | 59.6       | 69.3           | 61.1          | 55.8          | 62.1         | 49.9    |
> | **+ ours**       | –    | 32     | **76.9 (+5.6)** | **61.9 (+2.3)** | 70.6 | 62.7 | 54.8 | **62.7 (+0.6)** | **51.3 (+1.4)** |
>
> These results are on **long-video benchmarks—MLVU, VideoMME, LVBench**—covering topic reasoning, event ordering, counting, anomaly detection, etc. (**Fig. 15, 16 in the paper**).
>
> **VideoTree**[1] is a training-free agent that uses ChatGPT or other MLLMs to generate captions for many candidate segments and then queries ChatGPT to pick clips, which leads to dependence on ChatGPT and high inference cost.  **ViaRL**[2] uses Qwen2.5-VL-3B as a selector but requires **joint finetuning with Qwen2.5-VL-7B**, making it non–plug-and-play and expensive to train.
>
> By contrast, **K-frames is trained once as a standalone selector and is fully plug-and-play**, applicable to different downstream MLLMs without any modification.

---

> ### Author Response · Authors · 2025-11-25
> **Reply to all Reviewers**
>
> ### **Additional Benchmarks and Dataset Statistics**
>
> To highlight the focus on long-video understanding, we summarize the average video lengths of our benchmarks.
>
> **Table R2. Average video length of benchmarks (in seconds)**
>
> | Dataset | MLVU | VideoMME-S | VideoMME-M | VideoMME-L | VideoMME-Avg | LVBench | EgoSchema | NextQA |
> |---------|------|------------|------------|------------|--------------|---------|-----------|--------|
> | Length (s) | 930 | 80.7 | 515.9 | 2466.7 | 1024 | 473 | 180 | 44 |
>
> **Table R3. EgoSchema & NextQA (accuracy %)**
>
>
> | Method             | EgoSchema | NextQA        |
> |--------------------|:---------:|:-------------:|
> | MoReVQA            | –         | 69.2          |
> | LLoVi              | 61.2      | 67.7          |
> | VideoAgent         | 60.2      | 71.3          |
> | VideoAgent (variant)| 62.8     | –             |
> | LVNet              | 68.2      | 72.9          |
> | VideoTree          | 66.2      | 75.6          |
> | **K‑frames (ours)**| 73.2      | 83.7 |
>
> Our method is primarily designed for long-video understanding, which naturally contains multiple scenes and scene-level selection is crucial. For short videos, which often consist of only one single scene and visual content is limited, the need for sophisticated keyframe selection is less pronounced. EgoSchema and NextQA contain more egocentric and step-wise temporal reasoning questions which places higher demands on the downstream model’s video understanding capabilities. Our scene-driven strategy is inherently more effective on long-video benchmarks than on these shorter clips.
>
> ### **Inference-time Analysis**
>
> To answer the common concern about efficiency and overhead, we measure keyframe selection time on long videos:
>
> **Table R4. Inference time**
>
> | Method                      | Dataset   | Length (s) | Acc. | Inf. Time (s) |
> |-----------------------------|-----------|:----------:|:----:|:-------------:|
> | LLoVi                       | EgoSchema | 180        | 50.8 | –             |
> | LangRepo                    | EgoSchema | 180        | 60.8 | 87.2          |
> | VideoTree (Mistral‑7B)      | EgoSchema | 180        | 63.0 | 24.3          |
> | VideoTree (Mistral‑8×7B)    | EgoSchema | 180        | 71.0 | 50.3          |
> | **K‑frames (ours)**         | EgoSchema | 180        | 73.2 | 12.8          |
> | VideoTree* (Mistral‑7B)     | MLVU      | 930        | –    | >24.3         |
> | LanguageBind                | MLVU      | 930        | 52.3 | 11.2          |
> | **K‑frames (ours)**         | MLVU      | 930        | 60.4 | 10.6          |
>
> - On MLVU, K-frames **improves accuracy from 52.3 to 60.4** over LanguageBind[3] with **slightly reduced inference time (11.2 → 10.6s)**, demonstrating great performance in our proposed keyframe selector.
> - Compared with VideoTree[1], which depends on generating captions for many candidate shots and querying a large LLM, K-frames achieves competitive accuracy with **substantially less latency (e.g., 12.8s vs 24.3s on EgoSchema)**. The asterisk (*) denotes estimated time. Since caption generation accounts for a larger proportion of the total time in VideoTree, the MLVU dataset (930s)—whose length is significantly greater than that of EgoSchema (180s)—requires more time to generate additional captions to achieve optimal performance.
>
> [1] Xu Z, Dai Q, Xie T, et al. ViaRL: Adaptive Temporal Grounding via Visual Iterated Amplification Reinforcement Learning[ J]. arXiv preprint arXiv:2505.15447, 2025.
>
> [2] Wang Z, Yu S, Stengel-Eskin E, et al. Videotree: Adaptive tree-based video representation for llm reasoning on long videos[C]//Proceedings of the Computer Vision and Pattern Recognition Conference. 2025: 3272-3283.
>
> [3] Zhu, Bin, et al. "Languagebind: Extending video-language pretraining to n-modality by language-based semantic alignment." arXiv preprint arXiv:2310.01852, 2023.

---

### Meta-Review · Area_Chair_V1Ee · 2026-01-03

**Summary:**

The major  issues concerned by reviewers including (i) how distinct the contribution is from recent related work, (ii) whether the evaluation provides direct and convincing evidence of improved selection, and (iii) clarity and transparency issues around comparisons, runtime/overhead, RL description, and dataset construction choices. While the rebuttal addresses many concerns, questions about novelty relative to recent work, the strength/independence of evidence for selection quality, and dataset reliability remain only partially resolved.

**Reviewer Concerns:**

Addressed:
1. Missing comparisons and stronger baselines. Multiple reviewers asked for more direct comparisons, and the authors added new comparison tables to cover this.

2. Lack of direct evidence for clip/selection quality. One reviewer noted that downstream QA alone makes it hard to attribute gains to selection. The authors responded with a evaluation which provides a more direct proxy for selection quality, though it relies on the dataset’s own relevance annotations and no further comparison.

3. Overhead. Several reviewers asked about latency and the cost of the clip-finding stage. The authors provide a new inference-time study (Table R4), arguing K-frames is competitive with (or faster than) LanguageBind and substantially faster than VideoTree-like pipelines while improving accuracy.

4. Priority tags. A reviewer asked whether the “priority tag” mechanism is necessary. The authors added an ablation showing a weak but consistent benefit when weighting P1/P2 differently.

5. Broader tasks.  A reviewer questioned whether the claimed temporal-structure advantage shows up in temporal reasoning/action-centric settings. The authors argue existing long-video benchmarks already include such subtasks and report additional experiments.


Remaining issues
1. Novelty relative to recent work. The authors clarify that the approach is not graph-based and emphasize the “clip-first, any-k” framing as distinct from frame-level retrieval or subset-selection/RL-style selection. This resolves the factual mismatch in one review, but there may still be reasonable disagreement about how much the method advances beyond closely related recent query-aware, temporally structured selection approaches.

2. Dataset reliability and reproducibility. The authors provide a clearer description of the PeakClips annotation pipeline, report additional statistics, and state they will release prompts and scripts. That said, the annotations are still largely LLM-generated, and the discussion does not yet present strong evidence of large-scale human verification or annotation quality auditing.

3. Write-up clarity and documentation. While the rebuttal addresses many questions, at least one reviewer continued to flag clarity issues, especially around the dataset threshold choices (e.g., 4.3/4.9) and parts of the RL/GRPO description. The authors explain these thresholds were manually chosen and provide intuition, but the justification would be stronger with empirical sensitivity analysis or clearer documentation in the revised manuscript.

Note on Review by RZLy: This review seems to frame the method as graph-based, which the authors state is not the case, so some detailed critiques may not directly apply. However, its higher-level concerns (comparisons, efficiency/overhead, and dataset transparency) is reasonable and remain relevant.

**Reviewer Scores:**

Reviewer B5co may raise the score from 4 to 5 given that most concerns are addressed, but some points particularly the benefits of scene-based splitting, the significance of the temporal localization results, and the impact of the Priority Tag remain somewhat unclear/unsignificant.

Reviewer kpwF is likely to keep the score of 6, as the request for direct selection-quality evidence and overhead are covered in the rebuttal.

Reviewer RZLy is likely to keep the score of 4. Though authors state the review is based on a methodological misunderstanding, the broader concerns about novelty relative to recent work, evaluation strength, efficiency, and dataset transparency are reasonable and are only partially resolved in the rebuttal.

Reviewer LNvJ appreciated the rebuttal and kept follow-up concerns on hyper-param selection, likely to raise score from 4 to 5 .

---

### Decision · Program_Chairs · 2026-01-26

Reject